# Natural Policy Gradient Primal-Dual Method for Constrained Markov Decision Processes

**Dongsheng Ding**
ECE
University of Southern California
dongshed@usc.edu

**Kaiqing Zhang**
ECE and CSL
University of Illinois at Urbana-Champaign
kzhang66@illinois.edu

**Tamer Başar**
ECE and CSL
University of Illinois at Urbana-Champaign
basar1@illinois.edu

**Mihailo R. Jovanović**
ECE
University of Southern California
mihailo@usc.edu

## Abstract

We study sequential decision-making problems in which each agent aims to maximize the expected total reward while satisfying a constraint on the expected total utility. We employ the natural policy gradient method to solve the discounted infinite-horizon Constrained Markov Decision Processes (CMDPs) problem. Specifically, we propose a new Natural Policy Gradient Primal-Dual (NPG-PD) method for CMDPs which updates the primal variable via natural policy gradient ascent and the dual variable via projected sub-gradient descent. Even though the underlying maximization involves a nonconcave objective function and a nonconvex constraint set under the softmax policy parametrization, we prove that our method achieves global convergence with sublinear rates regarding both the optimality gap and the constraint violation. Such a convergence is independent of the size of the state-action space, i.e., it is dimension-free. Furthermore, for the general smooth policy class, we establish sublinear rates of convergence regarding both the optimality gap and the constraint violation, up to a function approximation error caused by restricted policy parametrization. Finally, we show that two sample-based NPG-PD algorithms inherit such non-asymptotic convergence properties and provide finite-sample complexity guarantees. To the best of our knowledge, our work is the first to establish non-asymptotic convergence guarantees of policy-based primal-dual methods for solving infinite-horizon discounted CMDPs. We also provide computational results to demonstrate merits of our approach.

## 1 Introduction

Reinforcement learning (RL) studies sequential decision-making problems where the agent aims to maximize its expected total reward by interacting with an unknown environment over time [44]. The model of Markov Decision Processes (MDPs) is usually used to represent the environment dynamics. However, in many safety-critical applications, e.g., in autonomous driving [19], robotics [35], cyber-security [58], and financial management [1], the agent is also subject to constraints on its utilities/costs. This naturally leads to a generalization of the environment dynamics to constrained MDPs (CMDPs) [4]. Besides maximizing the expected total reward, the agent also has to take into account the constraint on the expected total utility/cost as an additional learning objective.

Policy gradient (PG) methods [45], including the natural policy gradient (NPG) [21], have enjoyed substantial empirical success in solving MDPs [39, 24, 31, 40, 44]. PG methods, or more generally *direct policy search* methods, have also been used to solve CMDPs [47, 12, 11, 15, 46, 23, 36, 2, 43].

However, most existing theoretical guarantees are asymptotic in nature and/or only provide local convergence guarantees to stationary-point policies. Theoretical non-asymptotic global convergence guarantees are largely absent: for arbitrary initial points, algorithms with a finite number of iterations and samples converge to an $\epsilon$-optimal solution that enjoys $\epsilon$-optimality gap and $\epsilon$-constraint violation. It is thus imperative to establish theoretical guarantees for PG methods in solving CMDPs. Our motivation also comes from recent advances on the global convergence properties of PG methods [18, 56, 9, 48, 3, 57].

In this work, we provide a theoretical foundation for the non-asymptotic global convergence of the NPG method in solving CMDPs and answer the following questions: (i) can we employ NPG methods for solving CMDPs?; (ii) if and how fast do these methods converge to the globally optimal value within the underlying constraints?; (iii) what is the effect of the function approximation error caused by a restricted policy parametrization?; and (iv) what is the sample complexity of NPG methods?

**Contribution.** Our contribution is four-fold: (i) We propose a simple but effective primal-dual algorithm – Natural Policy Gradient Primal-Dual (NPG-PD) method – for solving discounted infinite-horizon CMDPs. We employ natural policy gradient ascent to update the primal variable and projected sub-gradient descent to update the dual variable; (ii) Even though we show that the maximization problem has a nonconcave objective function and nonconvex constraint set under the softmax policy parametrization, we prove that our NPG-PD method achieves global convergence with rate $O(1/\sqrt{T})$ regarding both the optimality gap and the constraint violation, where $T$ is the total number of iterations. Our convergence guarantees are dimension-free, i.e., the rate is independent of the size of the state-action space; (iii) For the general smooth policy class, we establish convergence with rate $O(1/\sqrt{T})$ for the optimality gap and $O(1/T^{1/4})$ for the constraint violation, up to a function approximation error caused by restricted policy parametrization; and (iv) We show that two sample-based NPG-PD algorithms that we propose inherit such non-asymptotic convergence properties and provide the finite-sample complexity guarantees. To the best of our knowledge, our work is the first to provide non-asymptotic convergence guarantees for solving infinite-horizon discounted CMDPs in the primal-dual framework.

**Related Work.** Our work is related to Lagrangian-based CMDP algorithms [4, 12, 11, 15, 46, 23, 37, 36, 53]. However, convergence guarantees of these algorithms are either local (to stationary-point or locally optimal policies) [11, 15, 46] or asymptotic [12]. When function approximation is used for policy parametrization, [53] recognized the lack of convexity and showed asymptotic convergence (to a stationary point) of a method based on successive convex relaxations. In contrast, we establish global convergence in spite of the lack of convexity. References [36, 37] are closely related to our work. In [37], the authors provide duality analysis for CMDPs in the policy space and propose a provably convergent dual descent algorithm by assuming access to a nonconvex optimization oracle. However, how to obtain the solution to this nonconvex optimization was not analyzed/understood, and the global convergence of their algorithm was not established. In [36], the authors provide a primal-dual algorithm but do not offer any theoretical justification. In spite of the lack of convexity, our work provides global convergence guarantees for a new primal-dual algorithm without using any optimization oracles. Other related policy optimization methods include CPG [47], CPO [2, 51], and IPPO [27]. However, theoretical guarantees for these algorithms are still lacking. Recently, optimism principles have been used for efficient exploration in CMDPs [42, 59, 16, 38, 17, 6]. In comparison, our work focuses on the optimization landscape within a primal-dual framework.

Our work is also pertinent to recent global convergence results for PG methods. References [18, 32, 33] provide global convergence guarantees for (natural) PG methods for nonconvex linear quadratic regulator problems in both discrete- and continuous-time. For general MDPs, [56] shows that locally optimal policies are achievable using PG methods with a simple reward-reshaping. Reference [48] shows that (natural) PG methods converge to the globally optimal value when overparametrized neural networks are used. As a variant of natural PG, trust-region policy optimization (TRPO) [39] has also been shown to converge to the globally optimal policy with overparametrized neural networks [26] and, in general, with regularized MDPs [41]. References [9, 10] study global optimality and convergence of PG methods from a policy iteration perspective. Reference [3] provides characterizations of the global convergence properties of (natural) PG methods regarding computational, approximation, and sample size issues. Recent advances along this line include references [30, 55, 13, 28]. While all these references handle a lack of convexity in the objective function, additional effort is required to deal with nonconvex constraint sets that arise in CMDPs, and our paper addresses this challenge.

## 2 Constrained Markov Decision Processes

Consider a discounted Constrained Markov Decision Process [4] – CMDP$(S, A, P, r, g, b, \gamma, \rho)$ – where $S$ is a finite state space, $A$ is a finite action space, $P$ is a transition probability measure which specifies the transition probability $P(s' \,|\, s, a)$ from state $s$ to the next state $s'$ under action $a \in A$, $r$: $S \times A \to [0, 1]$ is a reward function, $g$: $S \times A \to [0, 1]$ is a utility function, $b$ is a constraint offset, $\gamma \in [0, 1)$ is a discount factor, and $\rho$ is an initial state distribution over $S$.

A stochastic policy of an agent is a function $\pi$: $S \to \Delta_A$, determining a probability simplex $\Delta_A$ over action space $A$ chosen by the agent based on the current state, e.g., $a_t \sim \pi(\cdot \,|\, s_t)$ at time $t$ Let $\Pi$ be a set of all possible policies. A policy $\pi \in \Pi$, together with initial state distribution $\rho$, induces a distribution over trajectories $\tau = \{(s_t, a_t, r_t, g_t)\}_{t=0}^{\infty}$, where $s_0 \sim \rho$, $a_t \sim \pi(\cdot \,|\, s_t)$ and $s_{t+1} \sim P(\cdot \,|\, s_t, a_t)$ for all $t \geq 0$.

Given a policy $\pi$, the value functions $V_r^{\pi}, V_g^{\pi}$: $S \to \mathbb{R}$ associated with the reward $r$ or the utility $g$ are the following expected values of total rewards or utilities received under policy $\pi$, respectively,

$$V_r^{\pi}(s) := \mathbb{E}\left[\sum_{t=0}^{\infty} \gamma^t r(s_t, a_t) \,\big|\, \pi, s_0 = s\right] \text{ and } V_g^{\pi}(s) := \mathbb{E}\left[\sum_{t=0}^{\infty} \gamma^t g(s_t, a_t) \,\big|\, \pi, s_0 = s\right]$$

where the expectation $\mathbb{E}$ is taken over the randomness of the trajectory $\tau$ induced by $\pi$. We further introduce the state-action value functions $Q_r^{\pi}(s, a), Q_g^{\pi}(s, a)$: $S \times A \to \mathbb{R}$ when the agent starts from an arbitrary state-action pair $(s, a)$ and follows policy $\pi$, together with their advantage functions $A_r^{\pi}, A_g^{\pi}$: $S \times A \to \mathbb{R}$,

$$Q_{\diamond}^{\pi}(s, a) := \mathbb{E}\left[\sum_{t=0}^{\infty} \gamma^t \diamond (s_t, a_t) \,\big|\, \pi, s_0 = s, a_0 = a\right] \text{ and } A_{\diamond}^{\pi} := Q_{\diamond}^{\pi}(s, a) - V_{\diamond}^{\pi}(s)$$

where symbol $\diamond$ is $r$ or $g$. Since $r, g \in [0, 1]$, it is easy to see that $V_r^{\pi}(s), V_g^{\pi}(s) \in [0, 1/(1-\gamma)]$. Their expected values under $\rho$ are: $V_r^{\pi}(\rho) := \mathbb{E}_{s_0 \sim \rho}[V_r^{\pi}(s_0)]$ and $V_g^{\pi}(\rho) := \mathbb{E}_{s_0 \sim \rho}[V_g^{\pi}(s_0)]$.

Having defined policy, value/action-value functions for the discounted CMDP, the agent's goal is to find a policy that maximizes the expected reward value subject to a constraint on the expected utility value,

$$\underset{\pi \in \Pi}{\text{maximize}} \ V_r^{\pi}(\rho) \ \text{ subject to } \ V_g^{\pi}(\rho) \geq b \tag{1}$$

in which we maximize over all policies and we set constraint offset $b \in (0, 1/(1-\gamma)]$ to avoid triviality. For multiple constraints, our formulation (1) and convergence results are readily generalizable.

Via the method of Lagrange multipliers [8], we formulate the problem (1) into the following max-min problem for the associated Lagrangian $V_L^{\pi, \lambda}(\rho)$,

$$\underset{\pi \in \Pi}{\text{maximize}} \ \underset{\lambda \geq 0}{\text{minimize}} \ V_L^{\pi, \lambda}(\rho) := V_r^{\pi}(\rho) + \lambda \left(V_g^{\pi}(\rho) - b\right) \tag{2}$$

where $\pi$ is the primal variable and $\lambda$ is the nonnegative Lagrange multiplier or dual variable. The associated dual function is defined as $V_D^{\lambda}(\rho) := \text{maximize}_{\pi} V_L^{\pi, \lambda}(\rho)$.

Instead of the linear program method [4], this work focuses on *direct policy search* methods for solving the problem (2). Direct methods are attractive since they can deal with large state-action spaces via policy parameterization, e.g., neural nets, and they allow us to directly optimize/monitor the value functions that we are interested in. They are useful especially when we can utilize policy gradient estimates via simulations of the policy. It is worth mentioning that (1) is a nonconcave optimization problem as we prove in Lemma 3, and the decision is on an infinite-dimensional policy space $\Pi$. These reasons make the problem (1) challenging.

Nevertheless, the problem (1) has nice properties in the policy space once it is strictly feasible. We adapt the Slater condition in the constrained optimization [8] to assume strict feasibility of (1).

**Assumption 1** (Slater Condition). *There exists $\xi > 0$ and $\bar{\pi} \in \Pi$ such that $V_g^{\bar{\pi}}(\rho) - b \geq \xi$.*

The Slater condition is mild in practice since we usually have *a priori* knowledge on a strictly feasible policy, e.g., the minimal utility is achievable by a particular policy so that the constraint becomes loose. We will assume this throughout the paper.

Let an optimal primal variable be $\pi^{\star}$, i.e., an optimal solution to (1). Let an optimal dual variable be $\lambda^{\star} \in \text{argmin}_{\lambda \geq 0} V_D^{\lambda}(\rho)$. Use the shorthand notation $V_r^{\pi^{\star}}(\rho) = V_r^{\star}(\rho)$ and $V_D^{\lambda^{\star}}(\rho) = V_D^{\star}(\rho)$

whenever it is clear from the context. We now recall the strong duality for CMDPs [4, 36] and we further prove boundedness of optimal dual variable $\lambda^\star$ in Corollary 1 in Appendix C.

**Lemma 1** (Strong Duality and Boundedness of $\lambda^\star$)**.** *Let Assumption 1 hold. Then, (i)* $V_r^\star(\rho) = V_D^\star(\rho)$*; (ii)* $0 \leq \lambda^\star \leq (V_r^\star(\rho) - V_r^{\bar{\pi}}(\rho))/\xi$*.*

Let $v(\tau) = \text{maximize}_{\pi \in \Pi}\{V_r^\pi(\rho) \,|\, V_g^\pi(\rho) \geq b + \tau\}$ be the value function associated with problem (1). Using concavity of $v(\tau)$ given by [36, Proposition 1], we provide a useful bound on the constraint violation in Lemma 2 and it is proved in Theorem 6 in Appendix C. Take $[x]_+ = \max(x, 0)$.

**Lemma 2** (Constraint Violation)**.** *Let Assumption 1 hold. For any* $C \geq 2\lambda^\star$*, if there exists a policy* $\pi \in \Pi$ *and* $\delta > 0$ *such that* $V_r^\star(\rho) - V_r^\pi(\rho) + C[b - V_g^\pi(\rho)]_+ \leq \delta$*, then* $[b - V_g^\pi(\rho)]_+ \leq 2\delta/C$*.*

Aided by the above properties implied by the Slater condition, this work directly studies the max-min problem (2) in the primal-dual domain.

# 3 Policy Parametrization and Natural Policy Gradient Primal-Dual Method

To make problem (1) tractable, we introduce a set of parametrized policies $\{\pi_\theta \,|\, \theta \in \Theta\}$ where $\Theta$ is a finite-dimensional parameter space. We reduce the problem (1) into a parametric optimization problem,

$$\underset{\theta \in \Theta}{\text{maximize}} \;\; V_r^{\pi_\theta}(\rho) \;\; \text{subject to} \;\; V_g^{\pi_\theta}(\rho) \;\geq\; b \tag{3}$$

together with a parametric max-min problem (2) for the Lagrangian $V_L^{\pi_\theta, \lambda}(\rho)$,

$$\underset{\theta \in \Theta}{\text{maximize}} \;\; \underset{\lambda \geq 0}{\text{minimize}} \;\; V_L^{\pi_\theta, \lambda}(\rho) \;:=\; V_r^{\pi_\theta}(\rho) \;+\; \lambda\,(V_g^{\pi_\theta}(\rho) - b). \tag{4}$$

The dual function reads $V_D^\lambda(\rho) := \text{maximize}_\theta\, V_L^{\pi_\theta, \lambda}(\rho)$. The problem (3) is finite-dimensional, but still nonconcave even if the constraint is absent [3]. We state it formally as follows and prove it in Appendix A via an easily-constructed CMDP example.

**Lemma 3.** *There is a CMDP such that for the problem* (3)*,* $V_r^{\pi_\theta}(s)$ *is not concave and the constraint set* $\{\theta \in \Theta \,|\, V_g^{\pi_\theta}(s) \geq b\}$ *is not convex.*

The associated Lagrangian $V_L^{\pi_\theta, \lambda}(\rho)$ is thus nonconcave in $\theta$ and convex in $\lambda$ and the problem (4) is a nonconcave-convex max-min problem. Many algorithms, e.g., [25, 34, 50], for solving max-min optimization problems though, strong assumptions on the max-min structure or only stationary-point convergence guarantees make them not suitable here. In this work, we will exploit our problem geometry to propose a new method to study the max-min problem (4). Before doing that, we first introduce two classes of policies that we are interested in.

**Softmax Parametrization**. A natural class of policies is parametrized by the softmax function,

$$\pi_\theta(a \,|\, s) \;=\; \frac{\exp(\theta_{s,a})}{\sum_{a' \in A} \exp(\theta_{s,a'})} \;\; \text{for all} \;\; \theta \in \mathbb{R}^{|S||A|}. \tag{5}$$

Nice analytical properties of the softmax policy include completeness and differentiability. It can represent any stochastic policy, and its closure contains all stationary policies. Other reasons for us to begin with this policy class are: (i) it equips the policy with a rich structure so that the natural PG update works like the classical multiplicative weights update in the online learning literature, e.g., [14]; (ii) it has served as lens to interpreting the function approximation error [3]. It is a warm-up for studying convergence properties of many RL algorithms [9, 3, 30, 13].

**General Parametrization**. A general class of stochastic policies is given by $\{\pi_\theta \,|\, \theta \in \Theta\}$ in which we assume $\Theta \subset \mathbb{R}^d$ without providing the structure of $\pi_\theta$. The parameter space has dimension $d$. This policy class covers a more practical setting using function approximation, e.g., (deep) neural networks [26, 48]. However, when we choose $d \ll |S||A|$, the policy class has a limited expressiveness, and it may not contain all stochastic policies, e.g., being *restricted*. With this in mind, it is reasonable for our theory to define global convergence up to some error caused by the restricted policy class.

**Natural Policy Gradient Primal-Dual (NPG-PD) Method**. To introduce our method, we first introduce some useful definitions. The discounted visitation distribution $d_{s_0}^\pi$ of a policy $\pi$ and its

expectation over initial distribution $\rho$ are given by,

$$d^{\pi}_{s_0}(s) \ = \ (1 - \gamma) \sum_{t=0}^{\infty} \gamma^t P^{\pi}(s_t = s \,|\, s_0) \ \text{ and } \ d^{\pi}_{\rho}(s) \ = \ \mathbb{E}_{s_0 \sim \rho}\left[d^{\pi}_{s_0}(s)\right] \tag{6}$$

where $P^{\pi}(s_t = s \,|\, s_0)$ is the probability of visiting state $s$ at time $t$ when the agent follows the policy $\pi$ with initial state $s_0$. When the parametrized policy $\pi_\theta$ is clear from the context, we use $V_r^{\theta}(\rho)$ instead of $V_r^{\pi_\theta}(\rho)$, and similarly for others. When $\pi_\theta(\cdot \,|\, s)$ is differentiable and it is in the probability simplex, i.e., $\pi_\theta \in \Delta_A^{|S|}$ for all $\theta$, the policy gradient (PG) of the Lagrangian (4) reads,

$$\nabla_\theta V_L^{\theta,\lambda}(s_0) \ = \ \frac{1}{1-\gamma} \mathbb{E}_{s_0 \sim d^{\pi_\theta}_{s_0}} \mathbb{E}_{a \sim \pi_\theta(\cdot \,|\, s)} \left[\nabla_\theta \log \pi_\theta(a \,|\, s) \cdot A_L^{\theta,\lambda}(s,a)\right]$$

which equals $\nabla_\theta V_r^{\theta}(s_0) + \lambda \nabla_\theta V_g^{\theta}(s_0)$ where $A_L^{\theta,\lambda}(s,a) := A_r^{\theta}(s,a) + \lambda A_g^{\theta}(s,a)$. The Fisher information matrix induced by $\pi_\theta$ is $F_\rho(\theta) := \mathbb{E}_{s \sim d^{\pi_\theta}_\rho} \mathbb{E}_{a \sim \pi_\theta(\cdot \,|\, s)} \left[\nabla_\theta \log \pi_\theta(a \,|\, s) \left(\nabla_\theta \log \pi_\theta(a \,|\, s)\right)^{\top}\right]$.

With this notion, we propose a new policy gradient type method – Natural Policy Gradient Primal-Dual (NPG-PD) method – for the problem (4),

$$\theta^{(t+1)} \ = \ \theta^{(t)} + \eta_1 \, F_\rho(\theta^{(t)})^{\dagger} \cdot \nabla_\theta V_L^{\theta^{(t)},\lambda^{(t)}}(\rho) \ \text{ and } \ \lambda^{(t+1)} \ = \ \mathcal{P}_\Lambda\big(\lambda^{(t)} - \eta_2 \big(V_g^{\theta^{(t)}}(\rho) - b\big)\big) \tag{7}$$

where $A^{\dagger}$ takes the Moore-Penrose inverse of matrix $A$, $\mathcal{P}_\Lambda(x)$ projects $x$ into the interval $\Lambda$ that will be specified later, and $\eta_1 > 0, \eta_2 > 0$ are constants. This method displays a first-principle design of primal-dual updates: (i) the primal update $\theta^{(t+1)}$ performs gradient ascent using the natural policy gradient: $F_\rho(\theta^{(t)})^{\dagger} \cdot \nabla_\theta V_L^{(t)}(\rho)$, which is the policy gradient of $V_L^{(t)}(\rho)$ in the geometry induced by Fisher information $F_\rho(\theta^{(t)})$; (ii) the dual update $\lambda^{(t+1)}$ works as projected sub-gradient descent by adding up constraint violation $b - V_g^{(t)}(\rho)$. We use the shorthand $V_g^{(t)}(\rho)$ instead of $V_g^{\theta^{(t)}}(\rho)$, and similarly for others.

In what follows, we first establish global convergence of the NPG-PD method (7) under the softmax parametrization in Section 4. We move to the general parametrization in Section 5 and show convergence of a generalized version of (7). In the end, we propose two model-free sample-based algorithms for implementing (7) and analyze their sample complexities. Before our analysis, it is useful to recall: $V_\diamond^{\pi}(s_0) - V_\diamond^{\pi'}(s_0) = \frac{1}{1-\gamma} \mathbb{E}_{s \sim d^{\pi}_{s_0}, a \sim \pi(\cdot \,|\, s)}[A_\diamond^{\pi'}(s,a)]$ for any two policies $\pi, \pi'$, and any state $s_0$, where the symbol $\diamond$ is $r$ or $g$. This is from the performance difference lemma [20, 3].

## 4    Softmax Parametrization: Dimension-free Global Convergence

We now study the NPG-PD method (7) under the softmax parametrization (5). Thanks to the completeness of the softmax policy class, the strong duality in Lemma 1 holds on the closure of the softmax policy class. We establish the global convergence with dimension-independent convergence rates, i.e., *dimension-free*, while the maximization problem (3) is nonconcave.

We first exploit the softmax policy structure to show that the NPG-PD update (7) enjoys a concise primal update given as follows. We provide a proof in Appendix B.

**Lemma 4.** *Let* $\Lambda = [0, 2/((1-\gamma)\xi)]$. *Further let* $A_L^{(t)}(s,a) := A_r^{(t)}(s,a) + \lambda^{(t)} A_g^{(t)}(s,a)$ *and* $Z^{(t)}(s) := \sum_{a \in A} \pi^{(t)}(a \,|\, s) \exp\big(\frac{\eta_1}{1-\gamma} A_L^{(t)}(s,a)\big)$. *Using parametrized policy* (5), (7) *equals to*

$$\theta_{s,a}^{(t+1)} \ = \ \theta_{s,a}^{(t)} + \frac{\eta_1}{1-\gamma} A_L^{(t)}(s,a) \quad or \quad \pi^{(t+1)}(a \,|\, s) \ = \ \pi^{(t)}(a \,|\, s) \frac{\exp\big(\frac{\eta_1}{1-\gamma} A_L^{(t)}(s,a)\big)}{Z^{(t)}(s)} \tag{8}$$
$$and \ \lambda^{(t+1)} \ = \ \mathcal{P}_\Lambda\big(\lambda^{(t)} - \eta_2 \big(V_g^{(t)}(\rho) - b\big)\big).$$

Due to the Moore-Penrose inverse of Fisher information, the updates (8) are free of the state distribution $d^{\pi^{(t)}}_\rho$ that appears in (7) through the policy gradient. The policy update imitates the multiplicative weights update that is recognized in the online linear optimization [14]. Here, the linear function is translated as an advantage function of the current policy at each iteration.

Next, we show the global convergence of the algorithm (8) regarding the optimality gap: $V_r^{\star}(\rho) - V_r^{(t)}(\rho)$ and the constraint violation: $b - V_g^{(t)}(\rho)$. We prove it in Appendix D.

**Theorem 1** (Global Convergence: Softmax Parametrization). *Let Assumption 1 hold for $\xi > 0$. Fix $T > 0$, $\rho \in \Delta_S$, $\theta^{(0)} = 0$, and $\lambda^{(0)} = 0$. If we choose $\eta_1 = 2 \log |A|$ and $\eta_2 = (1-\gamma)/\sqrt{T}$, then*

the iterates $\pi^{(t)}$ generated by the algorithm (8) satisfy,

$$\text{(Optimality gap)} \quad \frac{1}{T} \sum_{t=0}^{T-1} \left( V_r^\star(\rho) - V_r^{(t)}(\rho) \right) \leq \frac{4}{(1-\gamma)^2} \frac{1}{\sqrt{T}} \tag{9a}$$

$$\text{(Constraint violation)} \quad \left[ \frac{1}{T} \sum_{t=0}^{T-1} \left( b - V_g^{(t)}(\rho) \right) \right]_+ \leq \frac{1/\xi + 4\xi}{(1-\gamma)^2} \frac{1}{\sqrt{T}}. \tag{9b}$$

What we capture in Theorem 1 is that on the average the reward value function converges to the global optimal one and the constraint violation decays to zero. Putting it differently, to find an $\epsilon$-near-optimal value, e.g., $\epsilon$-optimality gap and $\epsilon$-constraint violation, the number of steps is $O(1/\epsilon^2)$ which is *independent of the sizes of the state space or the action space*. Although the maximization problem (3) is nonconcave, our bound $(\sqrt{T}, \sqrt{T})$ for the accumulative optimality gap/constraint violation is better than the classical one $(\sqrt{T}, T^{3/4})$ [29] and matches the rate [52] for solving online *convex* optimization with *convex* constraint sets.

Comparing to the unconstrained setting [3, Section 5.3], our proof needs additional efforts. As shown in Lemma 6 in Appendix D, the reward value function is coupled with the utility value function and neither of them enjoy monotonic improvement for the vanilla natural policy gradient method. Therefore, we must introduce a new line of analysis. We first establish the bounded average performance in Lemma 7 in Appendix D. It enables us to bound the optimality gap via drift analysis of the dual update. To deduce the constraint violation, it is tempting to use methods from the constrained convex optimization, e.g., [29, 52, 49, 54]. However, they are not satisfactory due to slow rate or the needed extra assumption. Instead, we establish that the constraint violation enjoys the same rate as the optimality gap under the strong duality in Lemma 2, although our problem (3) is nonconcave. To the best of our knowledge, this appears to be the first such result for a class of "nonconvex" problems.

Regarding the global convergence, our proof of Theorem 1 holds for arbitrary initializations and we use $\theta^{(0)} = 0$, $\lambda^{(0)} = 0$ just to ease the exposition. We use this simplification in the sequel.

## 5 General Parametrization: Convergence Rate and Optimality

In this section, we consider a general policy class $\{\pi_\theta \mid \theta \in \Theta\}$ in which $\Theta \subset \mathbb{R}^d$ is the parameter space. Let us consider a more general form for the update (7) with $\Lambda = [0, \infty)$,

$$\theta^{(t+1)} = \theta^{(t)} + \frac{\eta_1}{1-\gamma} w^{(t)} \quad \text{and} \quad \lambda^{(t+1)} = \mathcal{P}_\Lambda \left( \lambda^{(t)} - \eta_2 \left( V_g^{(t)}(\rho) - b \right) \right) \tag{10}$$

where $w^{(t)}/(1-\gamma)$ is either the exact natural policy gradient (NPG) or some (sample-based) approximation of it. Note that in this general parametrization setting, the strong duality in Lemma 1 does not necessarily hold [36]. Thus, our analysis in Section 4 does not apply. Let us first generalize NPG. For a distribution over state-action pair: $\nu \in \Delta_{S \times A}$, the compatible function approximation error [21] is given by $E^\nu(w; \theta, \lambda) := \mathbb{E}_{s,a \sim \nu}[(A_L^{\theta,\lambda}(s,a) - w \cdot \nabla_\theta \log \pi_\theta(a \mid s))^2]$ where $A_L^{\theta,\lambda}(s,a) := A_r^\theta(s,a) + \lambda A_g^\theta(s,a)$, and the minimal error is $E_\star^\nu(\theta, \lambda) := \min_w E^\nu(w; \theta, \lambda)$. We view the natural PG in (7) as a minimizer of $E^\nu(w; \theta, \lambda)$ once we take $\nu(s,a) = d_\rho^{\pi_\theta}(s)\pi_\theta(a \mid s)$,

$$(1-\gamma)F_\rho(\theta)^\dagger \cdot \nabla_\theta V_L^{\theta,\lambda}(\mu) \in \underset{w}{\operatorname{argmin}} \, E^\nu(w; \theta, \lambda)$$

which follows from the first-order optimality condition. If the minimizer has zero compatible function approximation error, we have already established the global convergence in Theorem 1 for the softmax parametrization. However, this is not the case for a general policy class, since it may not include all possible policies, e.g., if we take $d \ll |S||A|$ for the tabular CMDP. The intuition behind *compatibility* is that we can use any one of the minimizers of $E^\nu(w; \theta, \lambda)$ and it does not affect the convergence properties of algorithm; see discussions in [21, 45, 3]. Note that our compatible function approximation error defines over the advantage function for the Lagrangian, which is natural and different from the unconstrained case.

However, $\nu$ is an unknown state-action measure of a feasible comparison policy $\pi$. To relieve this issue, we introduce an exploratory initial distribution $\nu_0$ over states and actions. We define a state-action visitation distribution $\nu_{\nu_0}^\pi$ of a policy $\pi$ as $\nu_{\nu_0}^\pi(s,a) = (1-\gamma)\mathbb{E}_{(s_0,a_0) \sim \nu_0} \sum_{t=0}^\infty \gamma^t P^\pi(s_t = s, a_t = a \mid s_0, a_0)$ where $P^\pi(s_t = s, a_t = a \mid s_0, a_0)$ is the probability of visiting state-action $(s,a)$ following policy $\pi$ from initial state-action $(s_0, a_0)$. We unload

notation $\nu_{\nu_0}^{\pi^{(t)}}$ as $\nu^{(t)}$ if it is clear from the context. We now update $w^{(t)}$ in (10) in a general manner,

$$w^{(t)} \in \underset{w}{\arg\min}\, E^{\nu^{(t)}}(w; \theta^{(t)}, \lambda^{(t)}) := \mathbb{E}_{s,a \sim \nu^{(t)}}\big[\big(A_L^{\theta^{(t)},\lambda^{(t)}}(s,a) - w \cdot \nabla_\theta \log \pi_\theta(a\,|\,s)\big)^2\big] \quad (11)$$

in which we assume that the minimizer is computed exactly.

To establish convergence theory, we adopt the standard smoothness assumption [56, 3].

**Assumption 2** (Policy Smoothness). *For all $s \in S$, $a \in A$, $\log \pi_\theta(a\,|\,s)$ is a $\beta$-smooth of $\theta$, i.e.,*

$$\|\nabla_\theta \log \pi_\theta(a\,|\,s) \,-\, \nabla_{\theta'} \log \pi_\theta(a\,|\,s)\| \;\leq\; \beta\,\|\theta \,-\, \theta'\| \;\; \text{for all } \theta, \theta' \in \mathbb{R}^d.$$

One example that satisfies Assumption 2 is the linear softmax policy [3]. Thus, Assumption 2 strictly generalizes our previous result for the softmax parametrization (5).

**Theorem 2** (Convergence and Optimality: General Parametrization). *Let Assumptions 1 and 2 hold with a policy class $\{\pi_\theta \,|\, \theta \in \Theta\}$. Fix a state distribution $\rho$, a state-action distribution $\nu_0$, and $T > 0$. Let the best feasible policy be $\pi^\star = \pi_\theta^\star$. Define the induced state-action visitation measure under $\pi^\star$: $\nu^\star(s,a) = d_\rho^{\pi^\star}(s)\pi^\star(a\,|\,s)$. Suppose the iterates $\pi^{(t)}$ and $\lambda^{(t)}$ generated by the method (7) with a general primal update (11), $\theta^{(0)} = 0$, $\lambda^{(0)} = 0$, and $\eta_1 = \eta_2 = 1/\sqrt{T}$ satisfy $E_\star^{\nu^{(t)}}(\theta^{(t)}, \lambda^{(t)}) \leq \epsilon_{approx}$ and $\|w^{(t)}\| \leq W$ for all $0 \leq t < T$. Then,*

$$\text{(Optimality gap)} \quad \frac{1}{T}\sum_{t=0}^{T-1}\big(V_r^\star(\rho) - V_r^{(t)}(\rho)\big) \;\leq\; \frac{C_1}{(1-\gamma)^3}\frac{1}{\sqrt{T}} + \sqrt{\frac{\epsilon_{approx}}{(1-\gamma)^3}\left\|\frac{\nu^\star}{\nu_0}\right\|_\infty}$$

$$\text{(Constraint violation)} \quad \left[\frac{1}{T}\sum_{t=0}^{T-1}\big(b - V_g^{(t)}(\rho)\big)\right]_+ \;\leq\; \frac{C_2}{(1-\gamma)^2}\frac{1}{T^{1/4}} + \left(\frac{4\epsilon_{approx}}{T(1-\gamma)^3}\left\|\frac{\nu^\star}{\nu_0}\right\|_\infty\right)^{1/4}$$

*where $C_1 := 1 + \log|A| + \beta W^2$ and $C_2 := \sqrt{3 + 2\lambda^\star + 2\log|A| + \beta W^2}$.*

We provide a proof for Theorem 2 in Appendix E. The optimality gap and the constraint violation decays to zero up to the approximation error $\epsilon_{approx}$, an upper bound of the compatible error. Such an error is negligible in the constraint violation due to the factor $1/T^{1/4}$. Theorem 2 generalizes (7) via compatible function approximation that raises a state-action distribution mismatch. The distribution mismatch coefficient $\|\nu^\star/\nu_0\|_\infty$ carries exploration duty in the PG methods [9, 3, 41]. This coefficient can be made finite and small, with an explorative/random enough initial distribution $\nu_0$ [3]. When the error $\epsilon_{approx}$ is zero, the distribution mismatch effect disappears, and Theorem 2 is similar to Theorem 1. This is the case for the tabular softmax parametrization or the linear MDP case [3]. The overparametrized neural nets [48] also can lead to small $\epsilon_{approx}$. Moreover, instead of $\|\nu^\star/\nu_0\|_\infty$, it is straightforward to interpret Theorem 2 using the concept of transfer error [3].

Since the strong duality does not necessarily hold, we cannot utilize the previous method to control the constraint violation. Consequently, there is a slightly slower rate that is similar to [29, 22]. We exploit the boundedness of value functions and return to the unparametrized problem (1) via the worst-case analysis. The worst-case gap notwithstanding, we show that the constraint violation enjoys a sublinear rate, and the function approximation error $\epsilon_{approx}$ appears as decaying sublinearly.

## 6 Sample-Based NPG-PD Algorithms

We have assumed access to the exact natural policy gradient in Section 4 or the ability to exactly solve the minimization of the compatible function approximation error in Section 5. In these connections, Theorem 1 and Theorem 2 have established non-asymptotic convergence results. We now leverage our theoretical results to design sample-based algorithms using only empirical estimates, i.e., *model-free*.

We build on the general version of the NPG-PD method (10) to propose a sample-based NPG-PD algorithm with function approximation and $\Lambda = [0, \infty)$,

$$\theta^{(t+1)} \;=\; \theta^{(t)} + \frac{\eta_1}{1-\gamma}\widehat{w}^{(t)} \;\text{ and }\; \lambda^{(t+1)} \;=\; \mathcal{P}_\Lambda\big(\lambda^{(t)} - \eta_2\big(\widehat{V}_g^{(t)}(\rho) - b\big)\big) \quad (12)$$

where the gradient $\widehat{w}^{(t)}$ and the value function $\widehat{V}_g^{(t)}(\rho)$ are sample-based estimates. We display our algorithm as Algorithm 1 in Appendix F. At each time $t$, the CMDP environment is executed for $K$ rounds and it terminates with a probability $1 - \gamma$ at each round. For the population problem (11), we run SGD $K$ rounds: $w_{k+1} = w_k - \alpha G_k$, to estimate $\widehat{w}^{(t)} = K^{-1}\sum_{k=1}^{K} w_k$; see [5]. Here, we use

the following $G_k$ as an estimate of the population gradient $\nabla_w E^{\nu^{(t)}}(w; \theta^{(t)}, \lambda^{(t)})$,

$$G_k = \left( w_k \cdot \nabla_\theta \log \pi^{(t)}(a \mid s) - \widehat{A}_L^{(t)}(s, a) \right) \cdot \nabla_\theta \log \pi^{(t)}(a \mid s)$$

where $\widehat{A}_L^{(t)}(s, a) = \widehat{Q}_L^{(t)}(s, a) - \widehat{V}_L^{(t)}(s)$; $\widehat{Q}_L^{(t)}(s, a)$ and $\widehat{V}_L^{(t)}(s)$ are undiscounted sums in each round. In addition, we run another $K$ rounds with initial $s \sim \rho$ to estimate $\widehat{V}_g^{(t)}(s)$ as an undiscounted sum in each round and take the average of $K$ rounds to obtain $\widehat{V}_g^{(t)}(\rho)$. As shown in Appendix F, $\widehat{Q}_L^{(t)}(s, a)$, $\widehat{V}_L^{(t)}(s)$, and $\widehat{V}_g^{(t)}(\rho)$ are unbiased.

To establish the convergence result, we make two assumptions that are standard in the literature [56, 3].

**Assumption 3** (Lipschitz Policy). *For $0 \leq t < T$, the policy $\pi^{(t)}$ satisfies $\left\| \nabla_\theta \log \pi^{(t)}(a \mid s) \right\| \leq L_\pi$.*

**Assumption 4** (Bounded Error and Weight). *Take $w^{(t)} = \mathrm{argmin}_w E^{\nu^{(t)}}(w; \theta^{(t)}, \lambda^{(t)})$. For $0 \leq t < T$, the iterates generated by Algorithm 1 satisfy,*

$$\mathbb{E}\left[ E_\star^{\nu^{(t)}}\left( \theta^{(t)}, \lambda^{(t)} \right) \right] \leq \epsilon_{approx}, \quad \mathbb{E}\left[ \|\widehat{w}^{(t)}\|^2 \right] \leq \widehat{W}^2, \quad and \quad \mathbb{E}\left[ \|w^{(t)}\|^2 \right] \leq W^2$$

*where the expectation is over randomness in $\theta^{(t)}$ and $\lambda^{(t)}$ in Algorithm 1.*

**Theorem 3** (Sample Complexity: General Parametrization). *Let Assumptions 1, 2, 3, and 4 hold with a policy class $\{\pi_\theta \mid \theta \in \Theta\}$. Fix a state distribution $\rho$, a state-action distribution $\nu_0$, and $T > 0$. Let the best feasible policy be $\pi^\star = \pi_{\theta^\star}$. Define the induced state-action visitation measure under $\pi^\star$: $\nu^\star(s, a) = d_\rho^{\pi^\star}(s) \pi^\star(a \mid s)$. Suppose the iterates $\pi^{(t)}$ and $\lambda^{(t)}$ are generated by the sample-based NPG-PD algorithm: Algorithm 1, with $\theta^{(0)} = 0$, $\lambda^{(0)} = 0$, $\eta_1 = \eta_2 = 1/\sqrt{T}$, and $\alpha = 1/L_\pi$, in which $K$ rounds of trajectory samples are used at each time $t$. Then,*

$$\mathbb{E}\left[ \frac{1}{T} \sum_{t=0}^{T-1} \left( V_r^\star(\rho) - V_r^{(t)}(\rho) \right) \right] \leq \frac{C_3}{(1-\gamma)^3} \frac{1}{\sqrt{T}} + \sqrt{\frac{1}{(1-\gamma)^3} \left\| \frac{\nu^\star}{\nu_0} \right\|_\infty} \left( \sqrt{\epsilon_{approx}} + \frac{C_5}{\sqrt{K}} \right)$$

$$\mathbb{E}\left[ \frac{1}{T} \sum_{t=0}^{T-1} \left( b - V_g^{(t)}(\rho) \right) \right]_+ \leq \frac{C_4}{(1-\gamma)^2} \frac{1}{T^{1/4}} + \left( \frac{4}{T(1-\gamma)^3} \left\| \frac{\nu^\star}{\nu_0} \right\|_\infty \right)^{1/4} \left( (\epsilon_{approx})^{1/4} + \frac{\sqrt{C_5}}{K^{1/4}} \right)$$

*where $C_3 := 2 + \log |A| + \beta \widehat{W}^2$, $C_4 := 2\sqrt{1 + \lambda^\star + C_3}$, and $C_5 := 2\sqrt{d} \left( WL_\pi + 1/(1-\gamma) \right)$.*

In Appendix G, we provide a proof for Theorem 3. Theorem 3 describes both the role of function approximation and the sampling effect. As more samples are used, the optimality gap and the constraint violation behave similarly as Theorem 2. The constraint violation is less susceptible to both the function approximation error and the sampling estimation error than the optimality gap.

Moreover, a special case of (12) is a sample-based primal-dual algorithm with softmax parametrization if we take $\widehat{w}^{(t)} = \widehat{A}_L^{(t)}$ and $\Lambda = [0, 2/((1-\gamma)\xi)]$. We describe our algorithm as Algorithm 2 in Appendix H and show its sample complexity; a proof is given in Appendix I.

**Theorem 4** (Sample Complexity: Softmax Parametrization). *Let Assumption 1 hold for $\xi > 0$. Fix a state distribution $\rho$ and $T > 0$. Suppose the iterates $\pi^{(t)}$ and $\lambda^{(t)}$ are generated by the sample-based NPG-PD algorithm: Algorithm 2, with $\theta^{(0)} = 0$, $\lambda^{(0)} = 0$, $\eta_1 = 2 \log |A|$ and $\eta_2 = (1-\gamma)/\sqrt{T}$, in which $K$ rounds of trajectory samples are used at each time $t$. Then,*

$$(\text{Optimality gap}) \quad \mathbb{E}\left[ \frac{1}{T} \sum_{t=0}^{T-1} \left( V_r^\star(\rho) - V_r^{(t)}(\rho) \right) \right] \leq \frac{5}{(1-\gamma)^2} \frac{1}{\sqrt{T}} + \frac{1}{(1-\gamma)K\sqrt{T}}$$

$$(\text{Constaint violation}) \quad \mathbb{E}\left[ \frac{1}{T} \sum_{t=0}^{T-1} \left( b - V_g^{(t)}(\rho) \right) \right]_+ \leq \frac{1/\xi + 5\xi}{(1-\gamma)^2} \frac{1}{\sqrt{T}} + \frac{\xi}{K\sqrt{T}}.$$

For the softmax parametrization, Theorem 4 shows better dependence on $T$ and $K$ than Theorem 3. If there is no sampling effect, the convergence rates match those in Theorem 1. It is noted that this result still has the property of being *dimension-free* for the optimality gap and the constraint violation.

To verify our convergence theory, we provide computational results by simulating the algorithm (8) and its sample-based version: Algorithm 2, for a finite CMDP with random initializations. Given $T > 0$, the total number of optimization iterations, our stepsizes in theorems become constants, and multiplying them with positive constants does not affect convergence rates. We generalize the shared MDP code [9] to CMDPs. We first compare the NPG-PD method (8) with the dualDescent [37] that

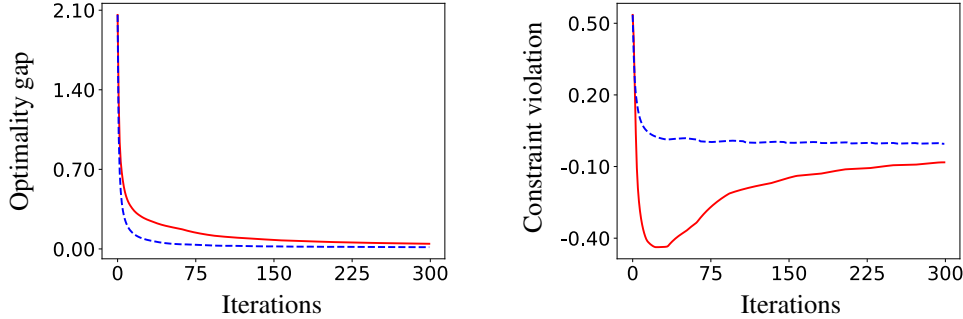

Figure 1: Comparison of the dualDescent [37] (- -) with the NPG-PD method (8) (—). In this experiment, we have randomly generated a CMDP with $|S| = 20$, $|A| = 10$, $\gamma = 0.8$, and $b = 3$, and chosen: $\eta_1 = \eta_2 = 1$.

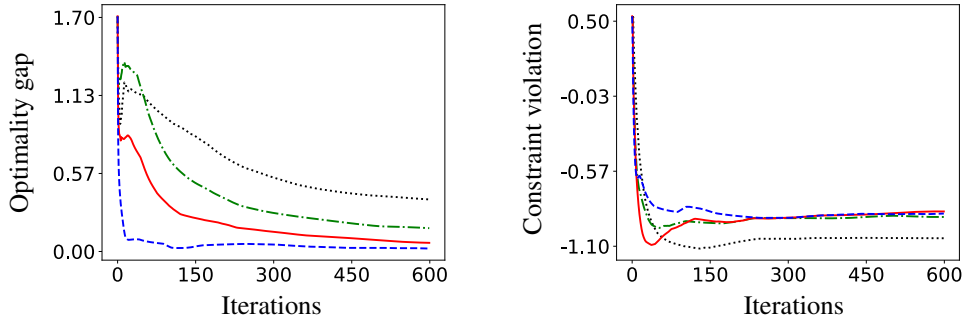

Figure 2: Comparison of the dualDescent [37] (- -) with the sample-based NPG-PD algorithm: Algorithm 2, using different sample sizes: $K = 20$ (····), $K = 50$ (-·-) and $K = 100$ (—). In this experiment, we have randomly generated a CMDP with $|S| = 20$, $|A| = 10$, $\gamma = 0.8$, and $b = 3$, and chosen: $\eta_1 = \eta_2 = 1$ for Algorithm 2, and $\eta = 1$, $K = 100$, and $L = 10$ for the dualDescent.

takes the exact PG method as an RL algorithm. In Figure 1, we see that both optimality gaps decay to zero quickly and our NPG-PD algorithm displays an outstanding constraint satisfaction. We also compare them by using only simulated policy gradients with the sample size $K$. In Figure 2, a key observation is that our sample-based NPG-PD algorithm performs as the dualDescent for large $K$. We point out that the dualDescent needs, roughly $L$ times more computation, than our algorithm since at each iteration it takes an extra inner-loop of executing an RL algorithm for $L$ steps. In this sense, our NPG-PD algorithm has better efficiency and is simple to apply without any inner-loop computation. See Appendix J for more experimental results.

## 7 Conclusion

In this paper, we have proposed an NPG-PD method for CMDPs with the primal natural policy gradient ascent and the dual projected sub-gradient descent. Even though the underlying maximization problem has a nonconcave objective function and a nonconvex constraint set, we provide a systematic study of the non-asymptotic convergence properties of this method with either the softmax parametrization or the general parametrization. We have also proposed two associated sample-based NPG-PD algorithms and established their finite-sample complexity guarantees. Our work is the first to offer non-asymptotic convergence guarantees of policy-based primal-dual methods for solving infinite-horizon discounted CMDPs.

A natural future direction is to investigate how we can achieve a fast rate, e.g., $O(1/T)$, for the NPG-PD method. The rate could be improved by utilizing the standard variance reduction technique. Another important direction is to study the generalization of our results to the vanilla primal-dual method without Fisher preconditioning. Moreover, it is relevant to exploit structure of particular CMDPs in order to provide improved convergence theory.

## Broader Impact

Our development could be added to a growing literature of constrained Markov decision processes (CMDPs) in a broad area of safe reinforcement learning (safe RL). Not only aiming to maximize the total reward, but almost all real-world sequential decision-making applications must also take control of safety regarding cost, utility, error rate, or efficiency, e.g., autonomous driving, medical test, financial management, and space exploration. Handling these additional safety objectives leads to constrained decision-making problems. Our research could be used to provide an algorithmic solution for practitioners to solve such constrained problems with non-asymptotic convergence and optimality guarantees. Our methodology could be new knowledge for RL researchers on the direct policy search methods for solving infinite-horizon discounted CMDPs.

The decision-making processes that build on our research could enjoy the flexibility of adding practical constraints and this would improve a large range of uses, e.g., autonomous systems, healthcare services, and financial and legal services. We may expect a broad range of societal implications and we list some of them as follows. The autonomous robotics could be deployed to hazard environments, e.g., forest fires or earthquakes, with added safety guarantees. This could accelerate rescuing while saving robotics. The discovery of medical treatments could be less risky by restraining the side effect. Thus the bias of treatments could be minimized effectively. The policymaker in government or enterprises could encourage economic productivity as much as they can but under law/environment/public health constraints. Overall, one could expect a lot of social welfare improvements supported by the uses of our research.

However, applying any theory to practice has to care about assumption/model mismatches. For example, our theory is in favor of well-defined feasible problems. This usually requires domain knowledge to justify. We would suggest domain experts develop guidelines for assumption/model validation. We would also encourage further work to establish the generalizability to other settings. Another issue could be the bias on gender or race. Policy parametrization selected by biased policymakers may inherit those biases. We would also encourage research to understand and mitigate the biases.

## Acknowledgments and Disclosure of Funding

D. Ding and M. R. Jovanović were supported by the National Science Foundation under Awards ECCS-1708906 and ECCS-1809833. Research of K. Zhang and T. Başar was supported in part by the US Army Research Laboratory (ARL) Cooperative Agreement W911NF-17-2-0196, and in part by the Office of Naval Research (ONR) MURI Grant N00014-16-1-2710.

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
