[Supplementary Material]

# Supplementary Materials for "Natural Policy Gradient Primal-Dual Method for Constrained Markov Decision Processes"

## A    Proof of Lemma 3

We prove Lemma 3 by providing a concrete CMDP example as shown in Figure 3. States $s_3$, $s_4$, and $s_5$ are terminal states with zero reward and utility. We consider non-trivial state $s_1$ with two actions: $a_1$ moving 'up' and $a_2$ going 'right', and the associated value functions are given by

$$V_r^\pi(s_1) = \pi(a_2 \mid s_1)\pi(a_1 \mid s_2)$$
$$V_g^\pi(s_1) = \pi(a_1 \mid s_1) + \pi(a_2 \mid s_1)\pi(a_1 \mid s_2).$$

Figure 3: An example of CMDP in the proof of Lemma 3 where $V_r^{\pi_\theta}(s)$ is nonconcave and the set $\{\theta \in \Theta \mid V_g^{\pi_\theta}(s) \geq b\}$ is not convex. The pair $(r, g)$ alongside the arrow depicts reward $r$ and utility $g$ of taking an action at certain state.

We consider the following two policies $\pi^{(1)}$ and $\pi^{(2)}$ using the softmax parametrization (5),

$$\theta^{(1)} = (\log 1, \log x, \log x, \log 1)$$
$$\theta^{(2)} = (-\log 1, -\log x, -\log x, -\log 1)$$

where the parameter takes form of $(\theta_{s_1,a_1}, \theta_{s_1,a_2}, \theta_{s_2,a_1}, \theta_{s_2,a_2})$ with $x > 0$.

First, we show that $V_r^\pi$ is not concave. We compute that

$$\pi^{(1)}(a_1 \mid s_1) = \frac{1}{1+x}, \ \pi^{(1)}(a_2 \mid s_1) = \frac{x}{1+x}, \ \pi^{(1)}(a_1 \mid s_2) = \frac{x}{1+x}$$

$$V_r^{(1)}(s_1) = \left(\frac{x}{1+x}\right)^2, \ V_g^{(1)}(s_1) = \frac{1+x+x^2}{(1+x)^2}$$

$$\pi^{(2)}(a_1 \mid s_1) = \frac{x}{1+x}, \ \pi^{(2)}(a_2 \mid s_1) = \frac{1}{1+x}, \ \pi^{(2)}(a_1 \mid s_2) = \frac{1}{1+x}$$

$$V_r^{(2)}(s_1) = \left(\frac{1}{1+x}\right)^2, \ V_g^{(2)}(s_1) = \frac{1+x+x^2}{(1+x)^2}.$$

Now, we consider policy $\pi^{(\zeta)}$,

$$\zeta\,\theta^{(1)} + (1-\zeta)\,\theta^{(2)} = \left(\log 1, \log\left(x^{2\zeta-1}\right), \log\left(x^{2\zeta-1}\right), \log 1\right)$$

for some $\zeta \in [0, 1]$, which is defined on the segment between $\theta^{(1)}$ and $\theta^{(2)}$. Therefore,

$$\pi^{(1)}(a_1 \mid s_1) = \frac{1}{1+x^{2\zeta-1}}, \ \pi^{(1)}(a_2 \mid s_1) = \frac{x^{2\zeta-1}}{1+x^{2\zeta-1}}, \ \pi^{(1)}(a_1 \mid s_2) = \frac{x^{2\zeta-1}}{1+x^{2\zeta-1}}$$

$$V_r^{(\zeta)}(s_1) = \left(\frac{x^{2\zeta-1}}{1+x^{2\zeta-1}}\right)^2, \ V_g^{(\zeta)}(s_1) = \frac{1+x^{2\zeta-1}+(x^{2\zeta-1})^2}{(1+x^{2\zeta-1})^2}.$$

When $x = 3$ and $\zeta = \frac{1}{2}$,

$$\frac{1}{2}V_r^{(1)}(s_1) + \frac{1}{2}V_r^{(2)}(s_1) = \frac{5}{16} > V_r^{(\frac{1}{2})}(s_1) = \frac{4}{16}$$

which implies that $V_r^\pi$ is not concave.

When $x = 10$ and $\zeta = \frac{1}{2}$,

$$V_g^{(1)}(s_1) \;=\; V_g^{(2)}(s_1) \;\geq\; 0.9 \;\text{ and }\; V_g^{(\frac{1}{2})}(s_1) \;=\; 0.75$$

which shows that if we take constraint offset $b = 0.9$, then $V_g^{(1)}(s_1) = V_g^{(2)}(s_1) \geq b$, and $V_g^{(\frac{1}{2})}(s_1) < b$ in which the policy $\pi^{(\frac{1}{2})}$ is infeasible. Therefore, the set $\{\theta \mid V_g^{\pi_\theta}(s) \geq b\}$ is not convex.

## B  Proof of Lemma 4

The dual update is based on Lemma 1. Since $\lambda^\star \leq \left(V_r^\star(\rho) - V_r^{\bar\pi}(\rho)\right)/\xi$ with $0 \leq V_r^\star, V_r^{\bar\pi} \leq \frac{1}{1-\gamma}$, we take projection interval $\Lambda = [0, \frac{2}{(1-\gamma)\xi}]$ such that upper bound $\frac{2}{(1-\gamma)\xi}$ is such that $\frac{2}{(1-\gamma)\xi} \geq 2\lambda^\star$.

We now verify the primal update. We expand the primal update in (7) into the following form,

$$\theta^{(t+1)} \;=\; \theta^{(t)} \;+\; \eta_1 F_\rho(\theta^{(t)})^\dagger \cdot \nabla_\theta V_r^{\theta^{(t)}}(\rho) \;+\; \eta_1 \lambda^{(t)} F_\rho(\theta^{(t)})^\dagger \cdot \nabla_\theta V_g^{\theta^{(t)}}(\rho). \tag{13}$$

We now deal with: $F_\rho(\theta^{(t)})^\dagger \cdot \nabla_\theta V_r^{\theta^{(t)}}(\rho)$ and $F_\rho(\theta^{(t)})^\dagger \cdot \nabla_\theta V_g^{\theta^{(t)}}(\rho)$. For the first one, the proof begins with solutions to the following approximation error minimization problem:

$$\underset{w \in \mathbb{R}^{|S||A|}}{\text{minimize}} \; E_r(w) \;:=\; \mathbb{E}_{s \sim d_\rho^{\pi_\theta}, a \sim \pi_\theta(a \mid s)} \left[ \left(A_r^{\pi_\theta}(s,a) - w \cdot \nabla_\theta \log \pi_\theta(a \mid s)\right)^2 \right].$$

Using the Moore-Penrose inverse, the optimal solution reads,

$$w_r^\star \;=\; F_\rho(\theta)^\dagger \mathbb{E}_{s \sim d_\rho^{\pi_\theta}, a \sim \pi_\theta(a \mid s)} \left[ \nabla_\theta \log \pi_\theta(a \mid s) A_r^{\pi_\theta, \lambda}(s,a) \right] \;=\; (1-\gamma) F_\rho(\theta)^\dagger \cdot \nabla_\theta V_r^{\pi_\theta, \lambda}(\rho)$$

where $F_\rho(\theta)$ is the Fisher information matrix induced by $\pi_\theta$. One key observation from this solution is that $w_r^\star$ is parallel to the natural PG direction $F_\rho(\theta)^\dagger \cdot \nabla_\theta V_r^{\pi_\theta, \lambda}(\rho)$.

On the other hand, it is easy to verify that $A_r^{\pi_\theta}$ is a minimizer of $E_r(w)$. The softmax parametrization (5) implies that

$$\frac{\partial \log \pi_\theta(a \mid s)}{\partial \theta_{s',a'}} \;=\; \mathbb{I}\{s = s'\}\left(\mathbb{I}\{a = a'\} - \pi_\theta(a' \mid s)\right) \tag{14}$$

where $\mathbb{I}\{E\}$ is the indicator function of event $E$ being true. Thus, we have

$$w \cdot \nabla_\theta \log \pi_\theta(a \mid s) \;=\; w_{s,a} \;-\; \sum_{a' \in A} w_{s,a'} \pi_\theta(a' \mid s).$$

The above equality together with the fact: $\sum_{a \in A} \pi_\theta(a \mid s) A_r^{\pi_\theta, \lambda}(s,a) = 0$, shows that $E_r(A_r^{\pi_\theta}) = 0$. However, $A_r^{\pi_\theta}$ may not be the unique minimizer. We consider the following general form of possible solutions,

$$A_r^{\pi_\theta} \;+\; u, \;\text{ where } \; u \;\in\; \mathbb{R}^{|S||A|}.$$

For any state $s$ and action $a$ such that $s$ is reachable under $\rho$, using (14) yields

$$u \cdot \nabla_\theta \log \pi_\theta(a \mid s) \;=\; u_{s,a} \;-\; \sum_{a' \in A} u_{s,a'} \pi_\theta(a' \mid s).$$

Here, we make use of the following fact: $\pi_\theta$ is a stochastic policy with $\pi_\theta(a \mid s) > 0$ for all actions $a$ in each state $s$, so that if a state is reachable under $\rho$, then it will also be reachable using $\pi_\theta$. Therefore, we require zero derivative at each reachable state:

$$u \cdot \nabla_\theta \log \pi_\theta(a \mid s) \;=\; 0$$

for all $s$, $a$ so that $u_{s,a}$ is independent of the action and becomes a constant $c_s$ for each $s$. Therefore, the minimizer of $E_r(w)$ is given up to some state-dependent offset,

$$F_\rho(\theta)^\dagger \cdot \nabla_\theta V_r^{\pi_\theta}(\rho) \;=\; \frac{A_r^{\pi_\theta}}{1-\gamma} \;+\; u \tag{15}$$

where $u_{s,a} = c_s$ for some $c_s \in \mathbb{R}$ for each state $s$ and action $a$.

We can repeat the above procedure for $F_\rho(\theta^{(t)})^\dagger \nabla_\theta V_g^{\theta^{(t)}}(\rho)$ and show,

$$F_\rho(\theta)^\dagger \cdot \nabla_\theta V_g^{\pi_\theta}(\rho) \;=\; \frac{A_g^{\pi_\theta}}{1-\gamma} \;+\; v \tag{16}$$

where $v_{s,a} = d_s$ for some $d_s \in \mathbb{R}$ for each state $s$ and action $a$.

Substituting (15) and (16) into the primal update (13) yields,

$$\theta^{(t+1)} = \theta^{(t)} + \frac{\eta_1}{1-\gamma} \left( A_r^{(t)} + \lambda^{(t)} A_g^{(t)} \right) + \eta_1 \left( u + \lambda^{(t)} v \right)$$

$$\pi^{(t+1)}(a \,|\, s) = \pi^{(t)}(a \,|\, s) \frac{\exp\left( \frac{\eta_1}{1-\gamma} \left( A_r^{(t)}(s,a) + \lambda^{(t)} A_g^{(t)}(s,a) \right) + \eta_1 \left( c_s + \lambda^{(t)} d_s \right) \right)}{Z^{(t)}(s)}$$

where the second equality also utilizes the normalization term $Z^{(t)}(s)$. Finally, we complete the proof by setting $c_s = d_s = 0$.

## C  Supporting Results from Optimization

We collect some optimization results from the literature for readers' convenience.

It is noted that all these results hold for the parametric setting of (3) and (4) if the parametrized policy class is *complete*, e.g., the closure of the softmax policy class (5). To rephrase them for our general purpose, we recall the maximization problem (1),

$$\underset{\pi \in \Pi}{\text{maximize}} \; V_r^\pi(\rho) \;\; \text{subject to} \;\; V_g^\pi(\rho) \geq b$$

in which we maximize over all policies and $b \in (0, 1/(1-\gamma))$ with $\gamma \in [0,1)$. Let the optimal solution be $\pi^\star$ such that

$$V_r^{\pi^\star}(\rho) = \underset{\pi \in \Pi}{\text{maximize}} \, \{ V_r^\pi(\rho) \,|\, V_g^\pi(\rho) \geq b \}.$$

Let the Lagrangian be $V_L^{\pi,\lambda}(\rho) := V_r^\pi(\rho) + \lambda(V_g^\pi(\rho) - b)$, where $\lambda \geq 0$ is the Lagrange multiplier or dual variable. The associated dual function is defined as

$$V_D^\lambda(\rho) := \underset{\pi \in \Pi}{\text{maximize}} \, V_L^{\pi,\lambda}(\rho) := V_r^\pi(\rho) + \lambda \left( V_g^\pi(\rho) - b \right)$$

and the optimal dual is $\lambda^\star = \mathrm{argmin}_{\lambda \geq 0} \, V_D^\lambda(\rho)$,

$$V_D^{\lambda^\star}(\rho) := \underset{\lambda \geq 0}{\text{minimize}} \, V_D^\lambda(\rho)$$

We recall that the problem (1) enjoys strong duality under the Slater condition [36, Proposition 1].

**Assumption 5** (Slater condition). *There exists $\xi > 0$ and $\bar{\pi}$ such that $V_g^{\bar{\pi}}(\rho) - b \geq \xi$.*

We use the shorthand notation $V_r^{\pi^\star}(\rho) = V_r^\star(\rho)$ and $V_D^{\lambda^\star}(\rho) = V_D^\star(\rho)$ whenever it is clear from the context.

**Lemma 5** (Strong duality). *[36, Proposition 1] If the Slater condition holds, then the strong duality holds,*

$$V_r^\star(\rho) = V_D^\star(\rho).$$

It is implied by the strong duality that the optimal solution to the dual problem: $\mathrm{minimize}_{\lambda \geq 0} \, V_D^\lambda(\rho)$ is obtained at $\lambda^\star$. Denote the set of all optimal dual variables as $\Lambda^\star$.

Under the Slater condition, a useful property of the dual variable is that the sublevel sets are bounded [7, Section 8.5]. Although our problem is nonconcave, we customize it as follows.

**Theorem 5** (Boundedness of sublevel sets of the dual function). *Let the Slater condition hold. Fix $C_\lambda \in \mathbb{R}$. For any $\lambda \in \{\lambda \geq 0 \,|\, V_D^\lambda(\rho) \leq C_\lambda\}$, it holds that*

$$\lambda \leq \frac{1}{\xi} \left( C_\lambda - V_r^{\bar{\pi}}(\rho) \right).$$

*Proof.* If $\lambda \in \{\lambda \geq 0 \,|\, V_D^\lambda(\rho) \leq C_\lambda\}$, then,

$$C_\lambda \geq V_D^\lambda(\rho) \geq V_r^{\bar{\pi}}(\rho) + \lambda \left( V_g^{\bar{\pi}}(\rho) - b \right) \geq V_r^{\bar{\pi}}(\rho) + \lambda \xi$$

where we utilize the Slater point $\bar{\pi}$ in the last inequality. We complete the proof by noting $\xi > 0$.  □

**Corollary 1.** *If we take $C_\lambda = V_r^\star(\rho) = V_D^\star$, then $\Lambda^\star = \{\lambda \geq 0 \,|\, V_D^\lambda(\rho) \leq C_\lambda\}$. Thus, for any $\lambda \in \Lambda^\star$,*

$$\lambda \leq \frac{1}{\xi} \left( V_r^{\pi^\star}(\rho) - V_r^{\bar{\pi}}(\rho) \right).$$

Another useful theorem from the convex optimization [7, Section 3.5] is given as follows. It describes that the constraint violation $b - V_g^\pi(\rho)$ can be bounded similarly even if we have some weak bound. We next state and prove it for our problem, which is used in our constraint violation analysis.

**Theorem 6.** *Let the Slater condition hold and $\lambda^\star \in \Lambda^\star$. Let $C_{\lambda^\star} \geq 2\lambda^\star$. Assume that $\widetilde{\pi} \in \Pi$ satisfies*

$$V_r^\star(\rho) - V_r^{\widetilde{\pi}}(\rho) + C_{\lambda^\star}\left[b - V_g^{\widetilde{\pi}}(\rho)\right]_+ \leq \delta.$$

*Then,*

$$\left[b - V_g^{\widetilde{\pi}}(\rho)\right]_+ \leq \frac{2\delta}{C_{\lambda^\star}}$$

*where $[x]_+ = \max(x, 0)$.*

*Proof.* Let

$$v(\tau) = \underset{\pi \in \Pi}{\text{maximize}} \left\{ V_r^\pi(\rho) \,|\, V_g^\pi(\rho) \geq b + \tau \right\}.$$

By the definition of $v(\tau)$, we have $v(0) = V_r^\star(\rho)$. We note the proof of [36, Proposition 1] that $v(\tau)$ is concave. First, we show that $-\lambda^\star \in \partial v(0)$. By the definition of Lagrangian $V_L^{\pi,\lambda}(\rho)$ and the strong duality,

$$V_L^{\pi,\lambda^\star}(\rho) \leq \underset{\pi \in \Pi}{\text{maximize}}\ V_L^{\pi,\lambda^\star}(\rho) = V_D^\star(\rho) = V_r^\star(\rho) = v(0), \quad \text{for all } \pi \in \Pi.$$

Hence, for any $\pi \in \{\pi \in \Pi \,|\, V_g^\pi(\rho) \geq b + \tau\}$, we have

$$\begin{aligned}
v(0) - \tau\lambda^\star &\geq V_L^{\pi,\lambda^\star}(\rho) - \tau\lambda^\star \\
&= V_r^\pi(\rho) + \lambda^\star(V_g^\pi(\rho) - b) - \tau\lambda^\star \\
&= V_r^\pi(\rho) + \lambda^\star(V_g^\pi(\rho) - b - \tau) \\
&\geq V_r^\pi(\rho).
\end{aligned}$$

If we maximize the right-hand side of above inequality over $\pi \in \{\pi \in \Pi \,|\, V_g^\pi(\rho) \geq b + \tau\}$, then

$$v(0) - \tau\lambda^\star \geq v(\tau) \tag{17}$$

which show that $-\lambda^\star \in \partial v(0)$.

On the other hand, if we take $\tau = \widetilde{\tau} := -(b - V_g^{\widetilde{\pi}}(\rho))_+$, then

$$V_r^{\widetilde{\pi}}(\rho) \leq V_r^\star(\rho) = v(0) \leq v(\widetilde{\tau}). \tag{18}$$

Combing (17) and (18) yields

$$V_r^{\widetilde{\pi}}(\rho) - V_r^\star(\rho) \leq -\widetilde{\tau}\lambda^\star.$$

Thus,

$$\begin{aligned}
(C_{\lambda^\star} - \lambda^\star)\,|\widetilde{\tau}| &= -\lambda^\star\,|\widetilde{\tau}| + C_{\lambda^\star}\,|\widetilde{\tau}| \\
&= \widetilde{\tau}\lambda^\star + C_{\lambda^\star}\,|\widetilde{\tau}| \\
&\leq V_r^\star(\rho) - V_r^{\widetilde{\pi}}(\rho) + C_{\lambda^\star}\,|\widetilde{\tau}|.
\end{aligned}$$

By our assumption and $\widetilde{\tau} = \left[b - V_g^{\widetilde{\pi}}(\rho)\right]_+$,

$$\left[b - V_g^{\widetilde{\pi}}(\rho)\right]_+ \leq \frac{\delta}{C_{\lambda^\star} - \lambda^\star} \leq \frac{2\delta}{C_{\lambda^\star}}.$$

$\square$

# D Proof of Theorem 1

We warm-up with an improvement lemma, stating a difference for two consecutive policies.

**Lemma 6** (Non-monotonic Improvement). *The iterates $\pi^{(t)}$ generated by algorithm (8) satisfy*

$$V_r^{(t+1)}(\mu) - V_r^{(t)}(\mu) + \lambda^{(t)}\left(V_g^{(t+1)}(\mu) - V_g^{(t)}(\mu)\right) \geq \frac{1-\gamma}{\eta_1} \mathbb{E}_{s \sim \mu} \log Z^{(t)}(s) \tag{19}$$

*and $\mathbb{E}_{s \sim \mu} \log Z^{(t)}(s) \geq 0$ for any initial state distributions $\mu$, where notation $d_\mu^{(t+1)}$ means $d_\mu^{\pi^{(t+1)}}$.*

*Proof.* To prove our main inequality, we first apply the performance difference lemma as follows:

$$
\begin{aligned}
V_r^{(t+1)}(\mu) - V_r^{(t)}(\mu) &= \frac{1}{1-\gamma} \mathbb{E}_{s \sim d_\mu^{(t+1)}, a \sim \pi^{(t+1)}(\cdot \mid s)} \left[ A_r^{(t)}(s,a) \right] \\
&= \frac{1}{1-\gamma} \mathbb{E}_{s \sim d_\mu^{(t+1)}} \left[ \sum_{a \in A} \pi^{(t+1)}(a \mid s) A_r^{(t)}(s,a) \right] \\
&= \frac{1}{\eta_1} \mathbb{E}_{s \sim d_\mu^{(t+1)}} \left[ \sum_{a \in A} \pi^{(t+1)}(a \mid s) \log \left( \frac{\pi^{(t+1)}(a \mid s)}{\pi^{(t)}(a \mid s)} Z^{(t)}(s) \right) \right] \\
&\quad - \frac{\lambda^{(t)}}{1-\gamma} \mathbb{E}_{s \sim d_\mu^{(t+1)}} \left[ \sum_{a \in A} \pi^{(t+1)}(a \mid s) A_g^{(t)}(s,a) \right] \\
&= \frac{1}{\eta_1} \mathbb{E}_{s \sim d_\mu^{(t+1)}} \left[ D_{\mathrm{KL}} \left( \pi^{(t+1)}(a \mid s) \,\|\, \pi^{(t)}(a \mid s) \right) \right] \\
&\quad + \frac{1}{\eta_1} \mathbb{E}_{s \sim d_\mu^{(t+1)}} \log Z^{(t)}(s) \\
&\quad - \frac{\lambda^{(t)}}{1-\gamma} \mathbb{E}_{s \sim d_\mu^{(t+1)}} \left[ \sum_{a \in A} \pi^{(t+1)}(a \mid s) A_g^{(t)}(s,a) \right] \\
&\geq \frac{1}{\eta_1} \mathbb{E}_{s \sim d_\mu^{(t+1)}} \log Z^{(t)}(s) \\
&\quad - \frac{\lambda^{(t)}}{1-\gamma} \mathbb{E}_{s \sim d_\mu^{(t+1)}} \left[ \sum_{a \in A} \pi^{(t+1)}(a \mid s) A_g^{(t)}(s,a) \right] \\
&= \frac{1}{\eta_1} \mathbb{E}_{s \sim d_\mu^{(t+1)}} \log Z^{(t)}(s) - \lambda^{(t)} \left( V_g^{(t+1)}(\mu) - V_g^{(t)}(\mu) \right)
\end{aligned}
$$

where the first two equalities are clear from definitions, the third equality is due to the multiplicative weights update in (8), the fourth equality utilizes the Kullback–Leibler divergence or relative entropy between two distributions $p, q$: $D_{\mathrm{KL}}(p \,\|\, q) := \mathbb{E}_{x \sim p} \log \frac{p(x)}{q(x)}$, we drop a nonnegative term in the inequality, and the last equality is due to the performance difference lemma again. Finally, we obtain the desired inequality by noting $d_\mu^{(t+1)} \geq (1-\gamma)\mu$ componentwise from (6).

It is easy to show that $\log Z^{(t)}(s) \geq 0$.

$$
\begin{aligned}
\log Z^{(t)}(s) &= \log \left( \sum_{a \in A} \pi^{(t)}(a \mid s) \exp \left( \frac{\eta_1}{1-\gamma} \left( A_r^{(t)}(s,a) + \lambda^{(t)} A_g^{(t)}(s,a) \right) \right) \right) \\
&\geq \sum_{a \in A} \pi^{(t)}(a \mid s) \log \left( \exp \left( \frac{\eta_1}{1-\gamma} \left( A_r^{(t)}(s,a) + \lambda^{(t)} A_g^{(t)}(s,a) \right) \right) \right) \\
&= \frac{\eta_1}{1-\gamma} \sum_{a \in A} \pi^{(t)}(a \mid s) \left( A_r^{(t)}(s,a) + \lambda^{(t)} A_g^{(t)}(s,a) \right) \\
&= \frac{\eta_1}{1-\gamma} \sum_{a \in A} \pi^{(t)}(a \mid s) A_r^{(t)}(s,a) + \frac{\eta_1}{1-\gamma} \lambda^{(t)} \sum_{a \in A} \pi^{(t)}(a \mid s) A_g^{(t)}(s,a) \\
&= 0
\end{aligned}
$$

In the above inequality, we apply the Jensen's inequality to the concave function $\log(x)$. The last equality is due to

$$
\sum_{a \in A} \pi^{(t)}(a \mid s) A_r^{(t)}(s,a) = \sum_{a \in A} \pi^{(t)}(a \mid s) A_g^{(t)}(s,a) = 0.
$$

$\square$

Next, we prove the average difference to the optimal policy.

**Lemma 7** (Bounded Average Performance). *Let Assumption 1 hold. Fix $T > 0$, $\rho \in \Delta_S$, $\theta^{(0)} = 0$, and $\lambda^{(0)} = 0$. Then the iterates $\pi^{(t)}$ and $\lambda^{(t)}$ generated by algorithm (8) satisfy*

$$
\frac{1}{T} \sum_{t=0}^{T-1} \left( V_r^\star(\rho) - V_r^{(t)}(\rho) \right) + \frac{1}{T} \sum_{t=0}^{T-1} \lambda^{(t)} \left( V_g^\star(\rho) - V_g^{(t)}(\rho) \right) \leq \frac{\log |A|}{\eta_1 T} + \frac{1}{(1-\gamma)^2 T} + \frac{2\eta_2}{(1-\gamma)^3}.
$$

*Proof.* Since $\rho$ is fixed, we unload notation $d_\rho^{\pi^\star}$ as $d^\star$. We first apply the performance difference lemma as follows:

$$
\begin{aligned}
V_r^\star(\rho) - V_r^{(t)}(\rho) &= \frac{1}{1-\gamma} \mathbb{E}_{s \sim d^\star} \left[ \sum_{a \in A} \pi^\star(a \mid s) A_r^{(t)}(s,a) \right] \\
&= \frac{1}{\eta_1} \mathbb{E}_{s \sim d^\star} \left[ \sum_{a \in A} \pi^\star(a \mid s) \log \left( \frac{\pi^{(t+1)}(a \mid s)}{\pi^{(t)}(a \mid s)} Z^{(t)}(s) \right) \right] \\
&\quad - \frac{\lambda^{(t)}}{1-\gamma} \mathbb{E}_{s \sim d^\star} \left[ \sum_{a \in A} \pi^\star(a \mid s) A_g^{(t)}(s,a) \right] \\
&= \frac{1}{\eta_1} \mathbb{E}_{s \sim d^\star} \left[ D_{\mathrm{KL}} \left( \pi^\star(a \mid s) \,\|\, \pi^{(t)}(a \mid s) \right) - D_{\mathrm{KL}} \left( \pi^\star(a \mid s) \,\|\, \pi^{(t+1)}(a \mid s) \right) \right] \\
&\quad + \frac{1}{\eta_1} \mathbb{E}_{s \sim d^\star} \log Z^{(t)}(s) - \frac{\lambda^{(t)}}{1-\gamma} \mathbb{E}_{s \sim d^\star} \left[ \sum_{a \in A} \pi^\star(a \mid s) A_g^{(t)}(s,a) \right] \\
&= \frac{1}{\eta_1} \mathbb{E}_{s \sim d^\star} \left[ D_{\mathrm{KL}} \left( \pi^\star(a \mid s) \,\|\, \pi^{(t)}(a \mid s) \right) - D_{\mathrm{KL}} \left( \pi^\star(a \mid s) \,\|\, \pi^{(t+1)}(a \mid s) \right) \right] \\
&\quad + \frac{1}{\eta_1} \mathbb{E}_{s \sim d^\star} \log Z^{(t)}(s) - \lambda^{(t)} \left( V_g^\star(\rho) - V_g^{(t)}(\rho) \right)
\end{aligned}
$$

$$(20)$$

where the second equality is due to the multiplicative weights update in (8), the third equality utilizes the Kullback–Leibler divergence or relative entropy between two distributions $p, q$: $D_{\mathrm{KL}}(p \,\|\, q) := \mathbb{E}_{x \sim p} \log \frac{p(x)}{q(x)}$, and the last equality is due to the performance difference lemma again.

According to Lemma 6, if we choose $\mu = d^\star$, then,

$$
V_r^{(t+1)}(d^\star) - V_r^{(t)}(d^\star) + \lambda^{(t)} \left( V_g^{(t+1)}(d^\star) - V_g^{(t)}(d^\star) \right) \geq \frac{1-\gamma}{\eta_1} \mathbb{E}_{s \sim d^\star} \log Z^{(t)}(s). \quad (21)
$$

Therefore, we have

$$
\begin{aligned}
&\frac{1}{T} \sum_{t=0}^{T-1} \left( V_r^\star(\rho) - V_r^{(t)}(\rho) \right) \\
&= \frac{1}{\eta_1 T} \sum_{t=0}^{T-1} \mathbb{E}_{s \sim d^\star} \left[ D_{\mathrm{KL}} \left( \pi^\star(a \mid s) \,\|\, \pi^{(t)}(a \mid s) \right) - D_{\mathrm{KL}} \left( \pi^\star(a \mid s) \,\|\, \pi^{(t+1)}(a \mid s) \right) \right] \\
&\quad + \frac{1}{\eta_1 T} \sum_{t=0}^{T-1} \mathbb{E}_{s \sim d^\star} \log Z^{(t)}(s) - \frac{1}{T} \sum_{t=0}^{T-1} \lambda^{(t)} \left( V_g^\star(\rho) - V_g^{(t)}(\rho) \right) \\
&\leq \frac{1}{\eta_1 T} \sum_{t=0}^{T-1} \mathbb{E}_{s \sim d^\star} \left[ D_{\mathrm{KL}} \left( \pi^\star(a \mid s) \,\|\, \pi^{(t)}(a \mid s) \right) - D_{\mathrm{KL}} \left( \pi^\star(a \mid s) \,\|\, \pi^{(t+1)}(a \mid s) \right) \right] \\
&\quad + \frac{1}{(1-\gamma)T} \sum_{t=0}^{T-1} \left( V_r^{(t+1)}(d^\star) - V_r^{(t)}(d^\star) \right) \\
&\quad + \frac{1}{(1-\gamma)T} \sum_{t=0}^{T-1} \lambda^{(t)} \left( V_g^{(t+1)}(d^\star) - V_g^{(t)}(d^\star) \right) - \frac{1}{T} \sum_{t=0}^{T-1} \lambda^{(t)} \left( V_g^\star(\rho) - V_g^{(t)}(\rho) \right) \\
&\leq \frac{1}{\eta_1 T} \mathbb{E}_{s \sim d^\star} D_{\mathrm{KL}} \left( \pi^\star(a \mid s) \,\|\, \pi^{(0)}(a \mid s) \right) + \frac{1}{(1-\gamma)T} V_r^{(T)}(d^\star) + \frac{2\eta_2}{(1-\gamma)^3} \\
&\quad - \frac{1}{T} \sum_{t=0}^{T-1} \lambda^{(t)} \left( V_g^\star(\rho) - V_g^{(t)}(\rho) \right)
\end{aligned}
$$

$$(22)$$

where in the second inequality we take telescoping sums for the first two sums and drop all non-positive terms: $D_{\mathrm{KL}} \left( \pi^\star(a \mid s) \,\|\, \pi^{(T)}(a \mid s) \right)$, $V_r^{(0)}(d^\star)$; we utilize the following result with $\mu = d^\star$

to the third sum,

$$\frac{1}{T} \sum_{t=0}^{T-1} \lambda^{(t)} \left( V_g^{(t+1)}(\mu) - V_g^{(t)}(\mu) \right)$$

$$= \frac{1}{T} \sum_{t=0}^{T-1} \left( \lambda^{(t+1)} V_g^{(t+1)}(\mu) - \lambda^{(t)} V_g^{(t)}(\mu) \right) + \frac{1}{T} \sum_{t=0}^{T-1} \left( \lambda^{(t)} - \lambda^{(t+1)} \right) V_g^{(t+1)}(\mu)$$

$$\leq \frac{1}{T} \lambda^{(T)} V_g^{(T)}(\mu) + \frac{1}{T} \sum_{t=0}^{T-1} \left| \lambda^{(t)} - \lambda^{(t+1)} \right| V_g^{(t+1)}(\mu)$$

$$\leq \frac{2\eta_2}{(1-\gamma)^2}$$

where in the first inequality we take telescoping sums for the first sum and drop a non-positive term, and in the second inequality we utilize the fact: $\left| \lambda^{(T)} \right| \leq \frac{\eta_2 T}{1-\gamma}$ and $\left| \lambda^{(t)} - \lambda^{(t+1)} \right| \leq \frac{\eta_2}{1-\gamma}$ due to the dual update in (8) and the non-expansiveness of projection, and the inequality $V_g^{(t)}(\mu) \leq \frac{1}{1-\gamma}$ due to the bounded value.

Finally, we use the fact that: $D_{\text{KL}}\left( p \,\|\, q \right) \leq \log |A|$, where $p, q \in \Delta_A$ and $q$ is the unifom distribution, $V_r^{(T)}(d^\star) \leq \frac{1}{1-\gamma}$, and $V_g^\star(\rho) \geq b$ to complete the proof. $\qquad\square$

We now prove our main statement in Theorem 1. We recall a key inequality from Lemma 7,

$$\frac{1}{T} \sum_{t=0}^{T-1} \left( V_r^\star(\rho) - V_r^{(t)}(\rho) \right) + \frac{1}{T} \sum_{t=0}^{T-1} \lambda^{(t)} \left( V_g^\star(\rho) - V_g^{(t)}(\rho) \right)$$

$$\leq \frac{\log |A|}{\eta_1 T} + \frac{1}{(1-\gamma)^2 T} + \frac{2\eta_2}{(1-\gamma)^3}. \tag{23}$$

**Bounding the optimality gap.** By the dual update in (8),

$$0 \leq \left( \lambda^{(T)} \right)^2 = \sum_{t=0}^{T-1} \left( (\lambda^{(t+1)})^2 - (\lambda^{(t)})^2 \right)$$

$$= \sum_{t=0}^{T-1} \left( \left( \mathcal{P}_\Lambda\left( 0, \lambda^{(t)} - \eta_2(V_g^{(t)}(\rho) - b) \right) \right)^2 - (\lambda^{(t)})^2 \right)$$

$$\leq \sum_{t=0}^{T-1} \left( \left( \lambda^{(t)} - \eta_2(V_g^{(t)}(\rho) - b) \right)^2 - (\lambda^{(t)})^2 \right) \tag{24}$$

$$= 2\eta_2 \sum_{t=0}^{T-1} \lambda^{(t)}(b - V_g^{(t)}(\rho)) + \eta_2^2 \sum_{t=0}^{T-1} (V_g^{(t)}(\rho) - b)^2$$

$$\leq 2\eta_2 \sum_{t=0}^{T-1} \lambda^{(t)} \left( V_g^\star(\rho) - V_g^{(t)}(\rho) \right) + \frac{\eta_2^2 T}{(1-\gamma)^2}$$

where the last inequality is due to the feasibility of the optimal policy $\pi^\star$: $V_g^\star(\rho) \geq b$, and $|V_g^{(t)}(\rho) - b| \leq \frac{1}{1-\gamma}$. The above inequality further implies,

$$-\frac{1}{T} \sum_{t=0}^{T-1} \lambda^{(t)} \left( V_g^\star(\rho) - V_g^{(t)}(\rho) \right) \leq \frac{\eta_2}{2(1-\gamma)^2}.$$

We substitute the above inequality into (23) and use the fact that: $D_{\text{KL}}\left( p \,\|\, q \right) \leq \log |A|$, where $p, q \in \Delta_A$ and $q$ is the uniform distribution to show the optimality gap bound, where we take $\eta_1 = 2 \log |A|$ and $\eta_2 = \frac{1-\gamma}{\sqrt{T}}$.

**Bounding the constraint violation**. By the dual update in (8), for any $\lambda \in \left[0, \frac{2}{(1-\gamma)\xi}\right]$,

$$
\begin{aligned}
|\lambda^{(t+1)} - \lambda|^2 &\leq \left|\lambda^{(t)} - \eta_2\left(V_g^{(t)}(\rho) - b\right) - \lambda\right|^2 \\
&= \left|\lambda^{(t)} - \lambda\right|^2 - 2\eta_2\left(V_g^{(t)}(\rho) - b\right)\left(\lambda^{(t)} - \lambda\right) + \eta_2^2\left(V_g^{(t)}(\rho) - b\right)^2 \\
&\leq \left|\lambda^{(t)} - \lambda\right|^2 - 2\eta_2\left(V_g^{(t)}(\rho) - b\right)\left(\lambda^{(t)} - \lambda\right) + \frac{\eta_2^2}{(1-\gamma)^2}
\end{aligned}
$$

where the first inequality is due to the non-expansiveness of projection operator $\mathcal{P}_\Lambda$ and the last inequality is due to $(V_g^{(t)}(\rho) - b)^2 \leq \frac{1}{(1-\gamma)^2}$. Summing it up from $t=0$ to $t=T-1$ and dividing it by $T$ yield

$$
0 \leq \frac{1}{T}|\lambda^{(T)} - \lambda|^2 \leq \frac{1}{T}\left|\lambda^{(0)} - \lambda\right|^2 - \frac{2\eta_2}{T}\sum_{t=0}^{T-1}\left(V_g^{(t)}(\rho) - b\right)\left(\lambda^{(t)} - \lambda\right) + \frac{\eta_2^2}{(1-\gamma)^2},
$$

which further implies,

$$
\frac{1}{T}\sum_{t=0}^{T-1}\left(V_g^{(t)}(\rho) - b\right)\left(\lambda^{(t)} - \lambda\right) \leq \frac{1}{2\eta_2 T}\left|\lambda^{(0)} - \lambda\right|^2 + \frac{\eta_2}{2(1-\gamma)^2}.
$$

We now add the above inequality into (23) and note $V_g^\star(\rho) \geq b$,

$$
\begin{aligned}
&\frac{1}{T}\sum_{t=0}^{T-1}\left(V_r^\star(\rho) - V_r^{(t)}(\rho)\right) + \frac{\lambda}{T}\sum_{t=0}^{T-1}\left(b - V_g^{(t)}(\rho)\right) \\
&\leq \frac{\log|A|}{\eta_1 T} + \frac{1}{(1-\gamma)^2 T} + \frac{2\eta_2}{(1-\gamma)^3} + \frac{1}{2\eta_2 T}\left|\lambda^{(0)} - \lambda\right|^2 + \frac{\eta_2}{2(1-\gamma)^2}.
\end{aligned}
$$

We take $\lambda = \frac{2}{(1-\gamma)\xi}$ when $\sum_{t=0}^{T-1}\left(b - V_g^{(t)}(\rho)\right) \geq 0$; otherwise $\lambda = 0$. Thus,

$$
\begin{aligned}
&V_r^\star(\rho) - \frac{1}{T}\sum_{t=0}^{T-1}V_r^{(t)}(\rho) + \frac{2}{(1-\gamma)\xi}\left[b - \frac{1}{T}\sum_{t=0}^{T-1}V_g^{(t)}(\rho)\right]_+ \\
&\leq \frac{\log|A|}{\eta_1 T} + \frac{1}{(1-\gamma)^2 T} + \frac{2\eta_2}{(1-\gamma)^3} + \frac{1}{2\eta_2(1-\gamma)^2\xi^2 T} + \frac{\eta_2}{2(1-\gamma)^2}.
\end{aligned}
$$

It should be mentioned that there exists a policy $\pi'$ such that $V_r^{\pi'}(\rho) = \frac{1}{T}\sum_{t=0}^{T-1}V_r^{(t)}(\rho)$ and $V_g^{\pi'}(\rho) = \frac{1}{T}\sum_{t=0}^{T-1}V_g^{(t)}(\rho)$. This holds in the following way (see [4, Chapter 10]). We recall that $V_r^{(t)}(\rho)$ and $V_g^{(t)}(\rho)$ are linear functions in the occupancy measure induced by policy $\pi^{(t)}$ and transition $P(s' \mid s, a)$. By the convexity of the set of occupancy measures, an average of $T$ occupancy measures is still an occupancy measure that produces a policy $\pi'$ with value $V_r^{\pi'}$ and $V_g^{\pi'}$.

Therefore,

$$
\begin{aligned}
&V_r^\star(\rho) - V_r^{\pi'}(\rho) + \frac{2}{(1-\gamma)\xi}\left[b - V_g^{\pi'}(\rho)\right]_+ \\
&\leq \frac{\log|A|}{\eta_1 T} + \frac{1}{(1-\gamma)^2 T} + \frac{2\eta_2}{(1-\gamma)^3} + \frac{1}{2\eta_2(1-\gamma)^2\xi^2 T} + \frac{\eta_2}{2(1-\gamma)^2}.
\end{aligned}
$$

We note that $\frac{2}{(1-\gamma)\xi} \geq 2\lambda^\star$. According to Lemma 2, we obtain

$$
\left[b - V_g^{\pi'}(\rho)\right]_+ \leq \frac{\xi\log|A|}{\eta_1 T} + \frac{\xi}{(1-\gamma)T} + \frac{2\eta_2\xi}{(1-\gamma)^2} + \frac{1}{2\eta_2(1-\gamma)\xi T} + \frac{\eta_2\xi}{2(1-\gamma)}
$$

which shows the constraint violation bound by noting $\frac{1}{T}\sum_{t=0}^{T-1}\left(b - V_g^{(t)}(\rho)\right) = b - V_g^{\pi'}(\rho)$ and taking $\eta_1 = 2\log|A|$ and $\eta_2 = \frac{1-\gamma}{\sqrt{T}}$.

# E   Proof of Theorem 2

We first characterize the effect of the compatible function approximation error on the convergence.

**Lemma 8.** *Let Assumption 1 and Assumption 2 hold for a policy class $\{\pi_\theta \mid \theta \in \Theta\}$. Fix a feasible comparison policy be $\pi$, a state distribution $\rho$, and $T > 0$. Define the induced state-action visitation measure under $\pi$: $\nu(s,a) = d_\rho^\pi(s)\pi(a \mid s)$. Suppose the iterates $\pi^{(t)}$ and $\lambda^{(t)}$ generated by the algorithm (10), $\theta^{(0)} = 0$, $\lambda^{(0)} = 0$, $\eta_1 = \eta_2 = 1/\sqrt{T}$ satisfy*

$$\frac{1}{T}\sum_{t=0}^{T-1} E^\nu(w^{(t)}; \theta^{(t)}, \lambda^{(t)}) \ \leq \ \widehat{\epsilon}_{approx} \ \text{ and } \ \|w^{(t)}\| \ \leq \ W \ \text{ for all } 0 \leq t < T.$$

*Then,*

*(Optimality gap)* $\quad \dfrac{1}{T}\sum_{t=0}^{T-1}\left(V_r^\pi(\rho) - V_r^{(t)}(\rho)\right) \ \leq \ \dfrac{C_1}{(1-\gamma)^3}\dfrac{1}{\sqrt{T}} + \dfrac{1}{1-\gamma}\sqrt{\widehat{\epsilon}_{approx}}$ $\quad$ (25a)

*(Constraint violation)* $\quad \left[\dfrac{1}{T}\sum_{t=0}^{T-1}\left(b - V_g^{(t)}(\rho)\right)\right]_+ \ \leq \ \dfrac{C_2}{(1-\gamma)^2}\dfrac{1}{T^{1/4}} + \dfrac{\sqrt{2}}{(1-\gamma)^{1/2}}\left(\dfrac{\widehat{\epsilon}_{approx}}{T}\right)^{1/4}$

$\qquad\qquad\qquad\qquad\qquad\qquad\qquad\qquad\qquad\qquad\qquad\qquad\qquad\qquad\qquad$ (25b)

*where $C_1 := 1 + \log|A| + \beta W^2$ and $C_2 := \sqrt{3 + 2\lambda^\star + 2\log|A| + \beta W^2}$.*

*Proof.* By Assumption 2, application of Taylor's theorem to $\log \pi^{(t)}(a \mid s)$ yields

$$\log\frac{\pi^{(t)}(a \mid s)}{\pi^{(t+1)}(a \mid s)} + \nabla_\theta \log \pi^{(t)}(a \mid s)\left(\theta^{(t+1)} - \theta^{(t)}\right) \ \leq \ \frac{\beta}{2}\|\theta^{(t+1)} - \theta^{(t)}\|^2$$

where $\theta^{(t+1)} - \theta^{(t)} = \frac{\eta_1}{1-\gamma}w^{(t)}$. We unload $d_\rho^\pi$ as $d$ since $\pi$ and $\rho$ are fixed. Therefore,

$\mathbb{E}_{s \sim d}\left(D_{\mathrm{KL}}(\pi(\cdot \mid s) \,\|\, \pi^{(t)}(\cdot \mid s)) - D_{\mathrm{KL}}(\pi(\cdot \mid s) \,\|\, \pi^{(t+1)}(\cdot \mid s))\right)$

$= \ -\mathbb{E}_{s \sim d}\mathbb{E}_{a \sim \pi(\cdot \mid s)}\log\dfrac{\pi^{(t)}(a \mid s)}{\pi^{(t+1)}(a \mid s)}$

$\geq \ \eta_1\mathbb{E}_{s \sim d}\mathbb{E}_{a \sim \pi(\cdot \mid s)}\left[\nabla_\theta \log \pi^{(t)}(a \mid s)\,w^{(t)}\right] - \beta\dfrac{\eta_1^2}{2(1-\gamma)^2}\|w^{(t)}\|^2$

$= \ \eta_1\mathbb{E}_{s \sim d}\mathbb{E}_{a \sim \pi(\cdot \mid s)}\left[\nabla_\theta \log \pi^{(t)}(a \mid s)\,w_r^{(t)}\right]$

$\qquad + \eta_1\lambda^{(t)}\mathbb{E}_{s \sim d}\mathbb{E}_{a \sim \pi(\cdot \mid s)}\left[\nabla_\theta \log \pi^{(t)}(a \mid s)\,w_g^{(t)}\right] - \beta\dfrac{\eta_1^2}{2(1-\gamma)^2}\|w^{(t)}\|^2$

$= \ \eta_1\mathbb{E}_{s \sim d}\mathbb{E}_{a \sim \pi(\cdot \mid s)}A_r^{(t)}(s,a) + \eta_1\lambda^{(t)}\mathbb{E}_{s \sim d}\mathbb{E}_{a \sim \pi(\cdot \mid s)}A_g^{(t)}(s,a)$

$\qquad + \eta_1\mathbb{E}_{s \sim d}\mathbb{E}_{a \sim \pi(\cdot \mid s)}\left[\nabla_\theta \log \pi^{(t)}(a \mid s)\left(w_r^{(t)} + \lambda^{(t)}w_g^{(t)}\right) - \left(A_r^{(t)}(s,a) + \lambda^{(t)}A_g^{(t)}(s,a)\right)\right]$

$\qquad - \beta\dfrac{\eta_1^2}{2(1-\gamma)^2}\|w^{(t)}\|^2$

$\geq \ \eta_1(1-\gamma)\left(V_r^\pi(\rho) - V_r^{(t)}(\rho)\right) + \eta_1(1-\gamma)\lambda^{(t)}\left(V_g^\pi(\rho) - V_g^{(t)}(\rho)\right)$

$\qquad - \eta_1\sqrt{\mathbb{E}_{s \sim d}\mathbb{E}_{a \sim \pi(\cdot \mid s)}\left[\left(\nabla_\theta \log \pi^{(t)}(a \mid s)\,w^{(t)} - A_L^{(t)}(s,a)\right)^2\right]}$

$\qquad - \beta\dfrac{\eta_1^2}{2(1-\gamma)^2}W^2$

where in the second equality we decompose $w^{(t)} = w_r^{(t)} + \lambda^{(t)}w_g^{(t)}$ for a given $\lambda^{(t)}$, in the last inequality we apply the performance difference lemma, the Jensen's inequality, and $\|w^{(t)}\| \leq W$.

Using notation of $E^\nu(w^{(t)}; \theta^{(t)}, \lambda^{(t)})$ and rearranging it yield,

$V_r^\pi(\rho) - V_r^{(t)}(\rho)$

$\leq \ \dfrac{1}{1-\gamma}\left(\dfrac{1}{\eta_1}\mathbb{E}_{s \sim d}\left(D_{\mathrm{KL}}(\pi(\cdot \mid s) \,\|\, \pi^{(t)}(\cdot \mid s)) - D_{\mathrm{KL}}(\pi(\cdot \mid s) \,\|\, \pi^{(t+1)}(\cdot \mid s))\right)\right)$

$\qquad + \dfrac{1}{1-\gamma}\sqrt{E^\nu(w^{(t)}; \theta^{(t)}, \lambda^{(t)})} + \beta\dfrac{\eta_1}{2(1-\gamma)^3}W^2 - \lambda^{(t)}\left(V_g^\pi(\rho) - V_g^{(t)}(\rho)\right)$

Therefore,

$$\frac{1}{T}\sum_{t=0}^{T-1}\left(V_r^\pi(\rho)-V_r^{(t)}(\rho)\right)$$

$$\leq \frac{1}{(1-\gamma)\eta_1 T}\sum_{t=0}^{T-1}\left(\mathbb{E}_{s\sim d}\left(D_{\mathrm{KL}}(\pi(\cdot\,|\,s)\,\|\,\pi^{(t)}(\cdot\,|\,s))-D_{\mathrm{KL}}(\pi(\cdot\,|\,s)\,\|\,\pi^{(t+1)}(\cdot\,|\,s))\right)\right)$$

$$+\frac{1}{(1-\gamma)T}\sum_{t=0}^{T-1}\sqrt{E^\nu(w^{(t)};\theta^{(t)},\lambda^{(t)})}+\beta\frac{\eta_1}{2(1-\gamma)}W^2-\frac{1}{T}\sum_{t=0}^{T-1}\lambda^{(t)}\left(V_g^\pi(\rho)-V_g^{(t)}(\rho)\right)$$

$$\leq \frac{\log|A|}{(1-\gamma)\eta_1 T}+\frac{1}{1-\gamma}\sqrt{\epsilon_{\mathrm{approx}}}+\beta\frac{\eta_1}{2(1-\gamma)^3}W^2+\frac{\eta_2}{2(1-\gamma)^2}$$

where in the last inequality we take telescoping sum of the first sum and drop a non-positive term; we apply the Jensen's inequality to $\sqrt{x}$ for the second sum and use the error bounds, and the last sum is due to,

$$0 \leq \left(\lambda^{(T)}\right)^2 = \sum_{t=0}^{T-1}\left((\lambda^{(t+1)})^2-(\lambda^{(t)})^2\right)$$

$$= \sum_{t=0}^{T-1}\left(\left(\max\left(0,\lambda^{(t)}-\eta_2(V_g^{(t)}(\rho)-b)\right)\right)^2-(\lambda^{(t)})^2\right)$$

$$\leq \sum_{t=0}^{T-1}\left(\left(\lambda^{(t)}-\eta_2(V_g^{(t)}(\rho)-b)\right)^2-(\lambda^{(t)})^2\right) \qquad (26)$$

$$= 2\eta_2\sum_{t=0}^{T-1}\lambda^{(t)}(b-V_g^{(t)}(\rho))+\eta_2^2\sum_{t=0}^{T-1}(V_g^{(t)}(\rho)-b)^2$$

$$\leq 2\eta_2\sum_{t=0}^{T-1}\lambda^{(t)}\left(V_g^\pi(\rho)-V_g^{(t)}(\rho)\right)+\frac{\eta_2^2 T}{(1-\gamma)^2}$$

where the last inequality is due to the feasibility of the comparison policy $\pi$: $V_g^\pi(\rho)\geq b$, and $|V_g^{(t)}(\rho)-b|\leq\frac{1}{1-\gamma}$. The above inequality further implies,

$$-\frac{1}{T}\sum_{t=0}^{T-1}\lambda^{(t)}\left(V_g^\pi(\rho)-V_g^{(t)}(\rho)\right) \leq \frac{\eta_2}{2(1-\gamma)^2}.$$

Now, we obtain the first bound by taking $\eta_1=\frac{1}{\sqrt{T}}$ and $\eta_2=\frac{1}{\sqrt{T}}$ and some simplification.

We next prove the second bound. By the dual update in (10), we have $\lambda^{(t+1)}-\lambda^{(t)}\geq-\eta_2(V_g^{(t)}(\rho)-b)$. Notice $\lambda^{(T)}\geq 0$. Therefore,

$$\left[\frac{1}{T}\sum_{t=0}^{T-1}(b-V_g^{(t)}(\rho))\right]_+ \leq \frac{1}{\eta_2 T}\sum_{t=0}^{T-1}\left(\lambda^{(t+1)}-\lambda^{(t)}\right) = \frac{1}{\eta_2 T}\lambda^{(T)}.$$

It comes down to establishing a bound on $\lambda^{(T)}$. Similar to (26), by the dual update in (10) with $\Lambda=[0,\infty)$,

$$0 \leq \left(\lambda^{(T)}\right)^2 \leq \sum_{t=0}^{T-1}\left(\left(\lambda^{(t)}-\eta_2(V_g^{(t)}(\rho)-b)\right)^2-(\lambda^{(t)})^2\right)$$

$$= 2\eta_2\sum_{t=0}^{T-1}\lambda^{(t)}(b-V_g^{(t)}(\rho))+\eta_2^2\sum_{t=0}^{T-1}(V_g^{(t)}(\rho)-b)^2$$

$$\leq 2\eta_2\sum_{t=0}^{T-1}\lambda^{(t)}\left(V_g^\pi(\rho)-V_g^{(t)}(\rho)\right)+\frac{\eta_2^2 T}{(1-\gamma)^2}$$

where the last inequality is due to the feasibility of the optimal policy $\pi^\pi$: $V_g^\pi(\rho) \geq b$, and $|V_g^{(t)}(\rho) - b| \leq \frac{1}{1-\gamma}$. Thus,

$$\left(\lambda^{(T)}\right)^2 \leq 2\eta_2 \sum_{t=0}^{T-1} \lambda^{(t)} \left(V_g^\pi(\rho) - V_g^{(t)}(\rho)\right) + \frac{\eta_2^2 T}{(1-\gamma)^2}.$$

Viewing the above bound, we return back to

$$\frac{1}{T} \sum_{t=0}^{T-1} \lambda^{(t)} \left(V_g^\pi(\rho) - V_g^{(t)}(\rho)\right)$$

$$\leq \frac{1}{(1-\gamma)\eta_1 T} \sum_{t=0}^{T-1} \left(\mathbb{E}_{s \sim d} \left(D_{\mathrm{KL}}(\pi(\cdot \mid s) \,\|\, \pi^{(t)}(\cdot \mid s)) - D_{\mathrm{KL}}(\pi(\cdot \mid s) \,\|\, \pi^{(t+1)}(\cdot \mid s))\right)\right)$$

$$+ \frac{1}{(1-\gamma)T} \sum_{t=0}^{T-1} \sqrt{E^\nu(w^{(t)}; \theta^{(t)}, \lambda^{(t)})} + \beta \frac{\eta_1}{2(1-\gamma)} W^2 - \frac{1}{T} \sum_{t=0}^{T-1} \left(V_r^\pi(\rho) - V_r^{(t)}(\rho)\right)$$

$$\leq \frac{\log |A|}{(1-\gamma)\eta_1 T} + \frac{1}{1-\gamma} \sqrt{\widehat{\epsilon}_{\mathrm{approx}}} + \beta \frac{\eta_1}{2(1-\gamma)^3} W^2 + \frac{\lambda^\star + 1}{1-\gamma}$$

where in the last inequality we take telescoping sum in the first sum and drop a non-positive term; we apply the Jensen's inequality to $\sqrt{x}$ for the second sum and use the error bounds, and the last sum is due to

$$
\begin{aligned}
V_r^\pi(\rho) &= V_r^\star(\rho) + (V_r^\pi(\rho) - V_r^\star(\rho)) \\
&\geq V_r^\star(\rho) - \frac{1}{1-\gamma} \\
&= V_D^\star(\rho) - \frac{1}{1-\gamma} \\
&= \underset{\pi}{\text{maximize}}\ V_r^\pi(\rho) + \lambda^\star \left(V_g^\pi(\rho) - b\right) - \frac{1}{1-\gamma} \\
&\geq V_r^{(t)}(\rho) + \lambda^\star \left(V_g^{(t)}(\rho) - b\right) - \frac{1}{1-\gamma} \\
&\geq V_r^{(t)}(\rho) - \frac{\lambda^\star + 1}{1-\gamma}
\end{aligned}
$$

where in the second equality we apply the strong duality in Lemma 1, the first and last inequalities are due to the boundedness of $|V_r^\pi(\rho) - V_r^\star(\rho)| \leq \frac{1}{1-\gamma}$ and $|V_g^{(t)}(\rho) - b| \leq \frac{1}{1-\gamma}$.

Therefore,

$$\frac{1}{\eta_2 T} \lambda^{(T)} \leq \frac{1}{\eta_2 T} \sqrt{2\eta_2 T \left(\frac{\log |A|}{(1-\gamma)\eta_1 T} + \frac{\sqrt{\widehat{\epsilon}_{\mathrm{approx}}}}{1-\gamma} + \frac{\eta_1 \beta W^2}{2(1-\gamma)^3} + \frac{\lambda^\star + 1}{1-\gamma}\right) + \frac{\eta_2^2 T}{(1-\gamma)^2}}$$

which leads to the desired bound by taking $\eta_1 = \frac{1}{\sqrt{T}}$ and $\eta_2 = \frac{1}{\sqrt{T}}$ and some simplification. $\qquad\square$

In Lemma 8, the compatible function approximation error shows up as an additive term in the upper bound for the optimality gap (25a) or the constraint violation (25b).

We now prove Theorem 2. It follows from the proof of Lemma 8 with an application of the inequality,

$$E^{\nu^\star}(w^{(t)}; \theta^{(t)}, \lambda^{(t)}) \leq \left\|\frac{\nu^\star}{\nu^{(t)}}\right\|_\infty E^{\nu^{(t)}}(w^{(t)}; \theta^{(t)}, \lambda^{(t)}) \leq \left\|\frac{\nu^\star}{\nu^{(t)}}\right\|_\infty \epsilon_{\mathrm{approx}} \leq \frac{\epsilon_{\mathrm{approx}}}{1-\gamma} \left\|\frac{\nu^\star}{\nu_0}\right\|_\infty$$

where the last inequality is due to $\nu^{(t)}(s, a) \geq (1-\gamma)\nu_0(s, a)$.

## F   Sample-Based NPG-PD Algorithm with Function Approximation

We describe a sample-based NPG-PD algorithm with function approximation in Algorithm 1. We note the computational complexity of Algorithm 1: each round has expected length $2/(1-\gamma)$ so the expected number of total samples is $4KT/(1-\gamma)$; the total number of gradient computations $\nabla_\theta \log \pi^{(t)}(a \mid s)$ is $2KT$; the total number of scalar multiplies, divides, and additions is $O(dKT + KT/(1-\gamma))$.

---

**Algorithm 1** Sample-based NPG-PD Algorithm with Function Approximation

---

1: **Initialization**: Learning rates $\eta_1$ and $\eta_2$, SGD learning rate $\alpha$, number of SGD iterations $K$, and simulation access to CMDP$(S, A, P, r, g, b, \gamma, \rho)$ under initial state-action distribution $\nu_0$.

2: **for** $t = 0, \ldots, T - 1$ **do**

3:     Initialize $\theta^{(0)} = 0$, $\lambda^{(0)} = 0$, $w_0 = 0$.

4:     **for** $k = 0, 1, \ldots, K - 1$ **do**

5:         Draw $(s, a) \sim \nu^{(t)}$.

6:         Execute policy $\pi^{(t)}$ starting from $(s, a)$ with a termination probability $1 - \gamma$ and estimate,

$$\widehat{Q}_L^{(t)}(s, a) = \sum_{k=0}^{K'-1} \left( r(s_k, a_k) + \lambda^{(t)} g(s_k, a_k) \right) \quad \text{where } s_0 = s, a_0 = a, K' \sim \text{Geo}(1-\gamma).$$

7:         Start from $s$, execute policy $\pi^{(t)}$ with a termination probability $1 - \gamma$ and estimate,

$$\widehat{V}_L^{(t)}(s) = \sum_{k=0}^{K'-1} \left( r(s_k, a_k) + \lambda^{(t)} g(s_k, a_k) \right) \quad \text{where } s_0 = s, K' \sim \text{Geo}(1 - \gamma).$$

8:         $\widehat{A}_L^{(t)}(s, a) = \widehat{Q}_L^{(t)}(s, a) - \widehat{V}_L^{(t)}(s)$.

9:         SGD update $w_{k+1} = w_k - \alpha \, G_k$, where

$$G_k = 2 \left( w_k \cdot \nabla_\theta \log \pi^{(t)}(a \,|\, s) - \widehat{A}_L^{(t)}(s, a) \right) \nabla_\theta \log \pi^{(t)}(a \,|\, s).$$

10:     **end for**

11:     Set $\widehat{w}^{(t)} = \frac{1}{K} \sum_{k=1}^{K} w_k$.

12:     Initialize $\widehat{V}_g^{(t)}(\rho) = 0$.

13:     **for** $k = 0, 1, \ldots, K - 1$ **do**

14:         Draw $s \sim \rho$ and draw $a \sim \pi^{(t)}(\cdot \,|\, s)$.

15:         Execute policy $\pi^{(t)}$ starting from $s$ with a termination probability $1 - \gamma$ and estimate,

$$\widehat{V}_g^{(t)}(s) = \sum_{k=0}^{K'-1} g(s_k, a_k) \quad \text{where } s_0 = s, a_0 = a, K' \sim \text{Geo}(1 - \gamma).$$

16:         Update $\widehat{V}_g^{(t)}(\rho) = \widehat{V}_g^{(t)}(\rho) + \frac{1}{K} \widehat{V}_g^{(t)}(s)$.

17:     **end for**

18:     Natural policy gradient primal-dual update

$$\begin{aligned} \theta^{(t+1)} &= \theta^{(t)} + \eta_1 \widehat{w}^{(t)} \\ \lambda^{(t+1)} &= \mathcal{P}_{[0,\infty)} \left( \lambda^{(t)} - \eta_2 \left( \widehat{V}_g^{(t)}(\rho) - b \right) \right). \end{aligned}$$

19: **end for**

---

We provide several unbiased estimates that are useful in our convergence proof.

$$
\begin{aligned}
\mathbb{E}\left[\widehat{V}_g^{(t)}(s)\right] &= \mathbb{E}\left[\sum_{k=0}^{K'-1} g(s_k, a_k) \,|\, \theta^{(t)}, s_0 = s\right] \\
&= \mathbb{E}\left[\sum_{k=0}^{\infty} \mathbb{I}\{K'-1 \geq k \geq 0\} g(s_k, a_k) \,|\, \theta^{(t)}, s_0 = s\right] \\
&= \sum_{k=0}^{\infty} \mathbb{E}\left[\mathbb{E}_{K'}\left[\mathbb{I}\{K'-1 \geq k \geq 0\}\right] g(s_k, a_k) \,|\, \theta^{(t)}, s_0 = s\right] \\
&= \sum_{k=0}^{\infty} \mathbb{E}\left[\gamma^k g(s_k, a_k) \,|\, \theta^{(t)}, s_0 = s\right] \\
&= \mathbb{E}\left[\sum_{k=0}^{\infty} \gamma^k g(s_k, a_k) \,|\, \theta^{(t)}, s_0 = s\right] \\
&= V_g^{(t)}(s)
\end{aligned}
$$

where we apply the Monotone Convergence Theorem and the Dominated Convergence Theorem for the third equality and the fifth equality to swap the expectation and the infinite sum, and in the fourth equality we use $\mathbb{E}_{K'}\left[\mathbb{I}\{K'-1 \geq k \geq 0\}\right] = 1 - P\left(K' < k\right) = \gamma^k$ since $K'$ follows a geometric distribution $\mathrm{Geo}(1-\gamma)$.

By a similar agument as above,

$$
\begin{aligned}
\mathbb{E}\left[\widehat{Q}_L^{(t)}(s, a)\right] &= \mathbb{E}\left[\sum_{k=0}^{K'-1} \left(r(s_k, a_k) + \lambda^{(t)} g(s_k, a_k)\right) \,|\, \theta^{(t)}, s_0 = s, a_0 = a\right] \\
&= \mathbb{E}\left[\sum_{k=0}^{K'-1} r(s_k, a_k) \,|\, \theta^{(t)}, s_0 = s, a_0 = a\right] \\
&\quad + \lambda^{(t)} \mathbb{E}\left[\sum_{k=0}^{K'-1} g(s_k, a_k) \,|\, \theta^{(t)}, s_0 = s, a_0 = a\right] \\
&= \mathbb{E}\left[\sum_{k=0}^{\infty} \mathbb{I}\{K'-1 \geq k \geq 0\} r(s_k, a_k) \,|\, \theta^{(t)}, s_0 = s, a_0 = a\right] \\
&\quad + \lambda^{(t)} \mathbb{E}\left[\sum_{k=0}^{\infty} \mathbb{I}\{K'-1 \geq k \geq 0\} g(s_k, a_k) \,|\, \theta^{(t)}, s_0 = s, a_0 = a\right] \\
&= \sum_{k=0}^{\infty} \mathbb{E}\left[\mathbb{E}_{K'}\left[\mathbb{I}\{K'-1 \geq k \geq 0\}\right] r(s_k, a_k) \,|\, \theta^{(t)}, s_0 = s, a_0 = a\right] \\
&\quad + \lambda^{(t)} \sum_{k=0}^{\infty} \mathbb{E}\left[\mathbb{E}_{K'}\left[\mathbb{I}\{K'-1 \geq k \geq 0\}\right] g(s_k, a_k) \,|\, \theta^{(t)}, s_0 = s, a_0 = a\right] \\
&= \sum_{k=0}^{\infty} \mathbb{E}\left[\gamma^k r(s_k, a_k) \,|\, \theta^{(t)}, s_0 = s, a_0 = a\right] \\
&\quad + \lambda^{(t)} \sum_{k=0}^{\infty} \mathbb{E}\left[\gamma^k g(s_k, a_k) \,|\, \theta^{(t)}, s_0 = s, a_0 = a\right] \\
&= \mathbb{E}\left[\sum_{k=0}^{\infty} \gamma^k r(s_k, a_k) \,|\, \theta^{(t)}, s_0 = s, a_0 = a\right] \\
&\quad + \lambda^{(t)} \mathbb{E}\left[\sum_{k=0}^{\infty} \gamma^k g(s_k, a_k) \,|\, \theta^{(t)}, s_0 = s, a_0 = a\right] \\
&= Q_r^{(t)}(s, a) + \lambda^{(t)} Q_g^{(t)}(s, a) \\
&= Q_L^{(t)}(s, a).
\end{aligned}
$$

Therefore,

$$\mathbb{E}\left[\widehat{A}_L^{(t)}(s,a)\right] = \mathbb{E}\left[\widehat{Q}_L^{(t)}(s,a)\right] - \mathbb{E}\left[\widehat{V}_L^{(t)}(s)\right] = Q_L^{(t)}(s,a) - V_L^{(t)}(s) = A_L^{(t)}(s,a).$$

We also provide a bound on the variance of $\widehat{V}_g^{(t)}(s)$.

$$\begin{aligned}
\text{Var}\left[\widehat{V}_g^{(t)}(s)\right] &= \mathbb{E}\left[\left(\widehat{V}_g^{(t)}(s) - V_g^{(t)}(s)\right)^2 \mid \theta^{(t)}, s_0 = s\right] \\
&= \mathbb{E}\left[\left(\sum_{k=0}^{K'-1} g(s_k, a_k) - V_g^{(t)}(s)\right)^2 \mid \theta^{(t)}, s_0 = s\right] \\
&= \mathbb{E}_{K'}\left[\mathbb{E}\left[\left(\sum_{k=0}^{K'-1} g(s_k, a_k) - V_g^{(t)}(s)\right)^2\right] \mid K'\right] \\
&\leq \mathbb{E}_{K'}\left[(K')^2 \mid K'\right] \\
&= \frac{1}{(1-\gamma)^2}
\end{aligned}$$

where the first inequality is due to $0 \leq g(x_k, a_k) \leq 1$ and $V_g^{(t)}(s) \geq 0$ and the last equality is clear from $K' \sim \text{Geo}(1-\gamma)$.

# G  Proof of Theorem 3

We split the proof into two parts. We state the roadmap here for readers' convenience. In the first part, we establish the following two bounds for the optimality gap and the constraint violation,

$$\mathbb{E}\left[\frac{1}{T}\sum_{t=0}^{T-1}\left(V_r^{\star}(\rho) - V_r^{(t)}(\rho)\right)\right] \leq \frac{\log|A|}{(1-\gamma)\eta_1 T} + \frac{1}{(1-\gamma)T}\sum_{t=0}^{T-1}\mathbb{E}\left[\sqrt{E^{\nu^{\star}}(\widehat{w}^{(t)};\theta^{(t)},\lambda^{(t)})}\right] \\
+ \beta\frac{\eta_1}{2(1-\gamma)^3}\widehat{W}^2 + \frac{2\eta_2}{(1-\gamma)^2} \tag{27}$$

and

$$\mathbb{E}\left[\frac{1}{T}\sum_{t=0}^{T-1}(b - V_g^{(t)}(\rho))\right]_+ \leq \\
\frac{1}{\eta_2 T}\sqrt{2\eta_2 T\left(\frac{\log|A|}{(1-\gamma)\eta_1 T} + \frac{1}{(1-\gamma)T}\sum_{t=0}^{T-1}\mathbb{E}\left[\sqrt{E^{\nu^{\star}}(\widehat{w}^{(t)};\theta^{(t)},\lambda^{(t)})}\right] + \frac{\eta_1\beta\widehat{W}^2}{2(1-\gamma)^3} + \frac{\lambda^{\star}+1}{1-\gamma}\right) + \frac{4\eta_2^2 T}{(1-\gamma)^2}}. \tag{28}$$

In the second part, we are seeking to control the error $\mathbb{E}\left[E^{\nu^{\star}}(\widehat{w}^{(t)};\theta^{(t)},\lambda^{(t)})\right]$,

$$\mathbb{E}\left[E^{\nu^{\star}}(\widehat{w}^{(t)};\theta^{(t)},\lambda^{(t)})\right] \leq \frac{1}{1-\gamma}\left\|\frac{\nu^{\star}}{\nu_0}\right\|_{\infty}\left(\epsilon_{\text{approx}} + \frac{2\left(\sqrt{d}WL_\pi + \frac{\sqrt{d}}{1-\gamma} + WL_\pi\right)^2}{K}\right). \tag{29}$$

Finally, we combine two parts to complete the proof by taking $\eta_1 = \frac{1}{\sqrt{T}}$ and $\eta_2 = \frac{1}{\sqrt{T}}$ and noting

$$\mathbb{E}\left[\sqrt{E^{\nu^{\star}}(\widehat{w}^{(t)};\theta^{(t)},\lambda^{(t)})}\right] \leq \sqrt{\mathbb{E}\left[E^{\nu^{\star}}(\widehat{w}^{(t)};\theta^{(t)},\lambda^{(t)})\right]}.$$

Let us begin with the first part. By Assumption 2, application of Taylor's theorem to $\log \pi^{(t)}(a \mid s)$ yields

$$\log\frac{\pi^{(t)}(a \mid s)}{\pi^{(t+1)}(a \mid s)} + \nabla_\theta \log \pi^{(t)}(a \mid s)\left(\theta^{(t+1)} - \theta^{(t)}\right) \leq \frac{\beta}{2}\|\theta^{(t+1)} - \theta^{(t)}\|^2$$

where $\theta^{(t+1)} - \theta^{(t)} = \frac{\eta_1}{1-\gamma}\widehat{w}^{(t)}$. We unload $d_\rho^{\pi^\star}$ as $d^\star$ since $\pi^\star$ and $\rho$ are fixed. Therefore,

$$\mathbb{E}_{s \sim d^\star} \left( D_{\mathrm{KL}}(\pi^\star(\cdot \mid s) \,\|\, \pi^{(t)}(\cdot \mid s)) - D_{\mathrm{KL}}(\pi^\star(\cdot \mid s) \,\|\, \pi^{(t+1)}(\cdot \mid s)) \right)$$

$$= \; -\mathbb{E}_{s \sim d^\star}\mathbb{E}_{a \sim \pi^\star(\cdot \mid s)} \log \frac{\pi^{(t)}(a \mid s)}{\pi^{(t+1)}(a \mid s)}$$

$$\geq \; \eta_1 \mathbb{E}_{s \sim d^\star}\mathbb{E}_{a \sim \pi^\star(\cdot \mid s)} \left[ \nabla_\theta \log \pi^{(t)}(a \mid s)\,\widehat{w}^{(t)} \right] - \beta \frac{\eta_1^2}{2(1-\gamma)^2} \|\widehat{w}^{(t)}\|^2$$

$$= \; \eta_1 \mathbb{E}_{s \sim d^\star}\mathbb{E}_{a \sim \pi^\star(\cdot \mid s)} \left[ \nabla_\theta \log \pi^{(t)}(a \mid s)\,\widehat{w}_r^{(t)} \right]$$
$$+ \; \eta_1 \lambda^{(t)} \mathbb{E}_{s \sim d^\star}\mathbb{E}_{a \sim \pi^\star(\cdot \mid s)} \left[ \nabla_\theta \log \pi^{(t)}(a \mid s)\,\widehat{w}_g^{(t)} \right]$$
$$- \; \beta \frac{\eta_1^2}{2(1-\gamma)^2} \|\widehat{w}^{(t)}\|^2$$

$$= \; \eta_1 \mathbb{E}_{s \sim d^\star}\mathbb{E}_{a \sim \pi^\star(\cdot \mid s)} A_r^{(t)}(s,a) \; + \; \eta_1 \lambda^{(t)} \mathbb{E}_{s \sim d^\star}\mathbb{E}_{a \sim \pi^\star(\cdot \mid s)} A_g^{(t)}(s,a)$$
$$+ \; \eta_1 \mathbb{E}_{s \sim d^\star}\mathbb{E}_{a \sim \pi^\star(\cdot \mid s)} \left[ \nabla_\theta \log \pi^{(t)}(a \mid s) \left( \widehat{w}_r^{(t)} + \lambda^{(t)}\widehat{w}_g^{(t)} \right) - \left( A_r^{(t)}(s,a) + \lambda^{(t)} A_g^{(t)}(s,a) \right) \right]$$
$$- \; \beta \frac{\eta_1^2}{2(1-\gamma)^2} \|\widehat{w}^{(t)}\|^2$$

$$\geq \; \eta_1 (1-\gamma) \left( V_r^\star(\rho) - V_r^{(t)}(\rho) \right) + \eta_1 (1-\gamma) \lambda^{(t)} \left( V_g^\star(\rho) - V_g^{(t)}(\rho) \right)$$
$$- \; \eta_1 \sqrt{ \mathbb{E}_{s \sim d^\star}\mathbb{E}_{a \sim \pi^\star(\cdot \mid s)} \left[ \left( \nabla_\theta \log \pi^{(t)}(a \mid s)\,\widehat{w}^{(t)} - A_L^{(t)}(s,a) \right)^2 \right] }$$
$$- \; \frac{\eta_1^2 \beta \widehat{W}^2}{2(1-\gamma)^2}$$

where in the second equality we decompose $\widehat{w}^{(t)} = \widehat{w}_r^{(t)} + \lambda^{(t)}\widehat{w}_g^{(t)}$ for a given $\lambda^{(t)}$, in the last inequality we apply the performance difference lemma, the Jensen's inequality, and $\|\widehat{w}^{(t)}\| \leq \widehat{W}$. Using notation of $E^{\nu^\star}(\widehat{w}^{(t)}; \theta^{(t)}, \lambda^{(t)})$ and rearranging it yields

$$V_r^\star(\rho) - V_r^{(t)}(\rho)$$

$$\leq \; \frac{1}{1-\gamma} \left( \frac{1}{\eta_1} \mathbb{E}_{s \sim d^\star} \left( D_{\mathrm{KL}}(\pi^\star(\cdot \mid s) \,\|\, \pi^{(t)}(\cdot \mid s)) - D_{\mathrm{KL}}(\pi^\star(\cdot \mid s) \,\|\, \pi^{(t+1)}(\cdot \mid s)) \right) \right)$$
$$+ \; \frac{1}{1-\gamma} \sqrt{E^{\nu^\star}(\widehat{w}^{(t)}; \theta^{(t)}, \lambda^{(t)})} + \beta \frac{\eta_1}{2(1-\gamma)^3} \widehat{W}^2 - \lambda^{(t)} \left( V_g^\star(\rho) - V_g^{(t)}(\rho) \right)$$

Therefore,

$$\frac{1}{T} \sum_{t=0}^{T-1} \left( V_r^\star(\rho) - V_r^{(t)}(\rho) \right)$$

$$\leq \; \frac{1}{(1-\gamma)\eta_1 T} \sum_{t=0}^{T-1} \left( \mathbb{E}_{s \sim d^\star} \left( D_{\mathrm{KL}}(\pi^\star(\cdot \mid s) \,\|\, \pi^{(t)}(\cdot \mid s)) - D_{\mathrm{KL}}(\pi^\star(\cdot \mid s) \,\|\, \pi^{(t+1)}(\cdot \mid s)) \right) \right)$$
$$+ \; \frac{1}{(1-\gamma)T} \sum_{t=0}^{T-1} \sqrt{E^{\nu^\star}(\widehat{w}^{(t)}; \theta^{(t)}, \lambda^{(t)})} + \frac{\eta_1 \beta \widehat{W}^2}{2(1-\gamma)} - \frac{1}{T} \sum_{t=0}^{T-1} \lambda^{(t)} \left( V_g^\star(\rho) - V_g^{(t)}(\rho) \right)$$

$$\leq \; \frac{\log |A|}{(1-\gamma)\eta_1 T} + \frac{1}{(1-\gamma)T} \sum_{t=0}^{T-1} \sqrt{E^{\nu^\star}(\widehat{w}^{(t)}; \theta^{(t)}, \lambda^{(t)})}$$
$$+ \; \frac{\eta_1 \beta \widehat{W}^2}{2(1-\gamma)^3} - \frac{1}{T} \sum_{t=0}^{T-1} \lambda^{(t)} \left( V_g^\star(\rho) - V_g^{(t)}(\rho) \right)$$

$$(30)$$

where in the last inequality we take telescoping sum of the first sum and drop a non-positive term. Taking expectation over randomness in $\theta^{(t)}$ and $\lambda^{(t)}$ yields

$$\mathbb{E}\left[\frac{1}{T}\sum_{t=0}^{T-1}\left(V_r^\star(\rho) - V_r^{(t)}(\rho)\right)\right] \;\leq\; \frac{\log|A|}{(1-\gamma)\eta_1 T} + \mathbb{E}\left[\frac{1}{(1-\gamma)T}\sum_{t=0}^{T-1}\sqrt{E^{\nu^\star}(\widehat{w}^{(t)};\theta^{(t)},\lambda^{(t)})}\right]$$
$$+ \frac{\eta_1\beta\widehat{W}^2}{2(1-\gamma)^3} - \mathbb{E}\left[\frac{1}{T}\sum_{t=0}^{T-1}\lambda^{(t)}\left(V_g^\star(\rho) - V_g^{(t)}(\rho)\right)\right]. \tag{31}$$

By the dual update in (12),

$$\begin{aligned}
0 \;\leq\; \left(\lambda^{(T)}\right)^2 &= \sum_{t=0}^{T-1}\left((\lambda^{(t+1)})^2 - (\lambda^{(t)})^2\right)\\
&= \sum_{t=0}^{T-1}\left(\left(\max\left(0, \lambda^{(t)} - \eta_2(\widehat{V}_g^{(t)}(\rho) - b)\right)\right)^2 - (\lambda^{(t)})^2\right)\\
&\leq \sum_{t=0}^{T-1}\left(\left(\lambda^{(t)} - \eta_2(\widehat{V}_g^{(t)}(\rho) - b)\right)^2 - (\lambda^{(t)})^2\right)\\
&= 2\eta_2\sum_{t=0}^{T-1}\lambda^{(t)}(b - \widehat{V}_g^{(t)}(\rho)) + \eta_2^2\sum_{t=0}^{T-1}(\widehat{V}_g^{(t)}(\rho) - b)^2\\
&\leq 2\eta_2\sum_{t=0}^{T-1}\lambda^{(t)}\left(V_g^\star(\rho) - V_g^{(t)}(\rho)\right) + 2\eta_2\sum_{t=0}^{T-1}\lambda^{(t)}\left(V_g^{(t)}(\rho) - \widehat{V}_g^{(t)}(\rho)\right)\\
&\quad + \eta_2^2\sum_{t=0}^{T-1}(\widehat{V}_g^{(t)}(\rho) - b)^2
\end{aligned} \tag{32}$$

where the last inequality is due to the feasibility of the policy $\pi^\star$: $V_g^\star(\rho) \geq b$. Since $V_g^{(t)}(\rho)$ is a population quantity and $\widehat{V}_g^{(t)}(\rho)$ is an estimate that is independent of $\lambda^{(t)}$ given $\theta^{(t-1)}$, $\lambda^{(t)}$ is independent of $V_g^{(t)}(\rho) - \widehat{V}_g^{(t)}(\rho)$ at time $t$ and thus $\mathbb{E}\left[\lambda^{(t)}\left(V_g^{(t)}(\rho) - \widehat{V}_g^{(t)}(\rho)\right)\right] = 0$ due to the fact $\mathbb{E}\left[\widehat{V}_g^{(t)}(\rho)\right] = V_g^{(t)}(\rho)$ (see Appendix F). Therefore,

$$\begin{aligned}
-\mathbb{E}\left[\frac{1}{T}\sum_{t=0}^{T-1}\lambda^{(t)}\left(V_g^\star(\rho) - V_g^{(t)}(\rho)\right)\right] &\leq \mathbb{E}\left[\frac{\eta_2}{2T}\sum_{t=0}^{T-1}(\widehat{V}_g^{(t)}(\rho) - b)^2\right]\\
&\leq \frac{\eta_2}{2(1-\gamma)^2}\left(1 + \frac{K+1}{K}\right)
\end{aligned}$$

where in the second inequality we drop a non-positive term and use the fact (see Appendix F),

$$\mathbb{E}\left[\widehat{V}_g^{(t)}(\rho)\right] \;=\; V_g^{(t)}(\rho)$$

and

$$\begin{aligned}
\mathbb{E}\left[\left(\widehat{V}_g^{(t)}(\rho)\right)^2\right] &= \frac{1}{K}\mathbb{E}\left[\left(\widehat{V}_g^{(t)}(s)\right)^2\right] + \frac{K-1}{K}\mathbb{E}\left[\widehat{V}_g^{(t)}(s)\right]\mathbb{E}\left[\widehat{V}_g^{(t)}(s)\right]\\
&= \frac{1}{K}\left(\mathrm{Var}\left[\widehat{V}_g^{(t)}(s)\right] + \left(\mathbb{E}\left[\widehat{V}_g^{(t)}(s)\right]\right)^2\right) + \frac{K-1}{K(1-\gamma)^2}\\
&= \frac{1}{K}\left(\mathrm{Var}\left[\widehat{V}_g^{(t)}(s)\right] + \left(V_g^{(t)}(s)\right)^2\right) + \frac{K-1}{K(1-\gamma)^2}\\
&\leq \frac{2}{K(1-\gamma)^2} + \frac{K-1}{K(1-\gamma)^2}
\end{aligned}$$

where the first equality is due to line 16 of Algorithm 1; the last inequality is due to $\mathrm{Var}\left[\widehat{V}_g^{(t)}(s)\right] \leq \frac{1}{(1-\gamma)^2}$ (see Appendix F) and $0 \leq V_g^{(t)}(s) \leq \frac{1}{1-\gamma}$.

We now return to (31) and apply $1 + \frac{K+1}{K} \leq 4$ to obtain (27).

On the other hand, by the dual update in (12), we have $\lambda^{(t+1)} - \lambda^{(t)} \geq -\eta_2(\widehat{V}_g^{(t)}(\rho) - b)$. Therefore,

$$
\begin{aligned}
\frac{1}{T}\sum_{t=0}^{T-1}(b - V_g^{(t)}(\rho)) &\leq \left[\frac{1}{T}\sum_{t=0}^{T-1}(b - \widehat{V}_g^{(t)}(\rho))\right]_+ + \frac{1}{T}\sum_{t=0}^{T-1}\left(\widehat{V}_g^{(t)}(\rho) - V_g^{(t)}(\rho)\right) \\
&\leq \frac{1}{\eta_2 T}\sum_{t=0}^{T-1}\left(\lambda^{(t+1)} - \lambda^{(t)}\right) + \frac{1}{T}\sum_{t=0}^{T-1}\left(\widehat{V}_g^{(t)}(\rho) - V_g^{(t)}(\rho)\right) \\
&= \frac{1}{\eta_2 T}\lambda^{(T)} + \frac{1}{T}\sum_{t=0}^{T-1}\left(\widehat{V}_g^{(t)}(\rho) - V_g^{(t)}(\rho)\right).
\end{aligned}
$$

By $\mathbb{E}\left[\widehat{V}_g^{(t)}(\rho)\right] = V_g^{(t)}(\rho)$ (see Appendix F),

$$
\mathbb{E}\left[\frac{1}{T}\sum_{t=0}^{T-1}(b - V_g^{(t)}(\rho))\right]_+ \leq \frac{1}{\eta_2 T}\mathbb{E}\left[\lambda^{(T)}\right] \tag{33}
$$

Also, by (32),

$$
\begin{aligned}
\mathbb{E}\left[\left(\lambda^{(T)}\right)^2\right] &\leq 2\eta_2\mathbb{E}\left[\sum_{t=0}^{T-1}\lambda^{(t)}\left(V_g^\star(\rho) - V_g^{(t)}(\rho)\right)\right] + 2\eta_2\mathbb{E}\left[\sum_{t=0}^{T-1}\lambda^{(t)}\left(V_g^{(t)}(\rho) - \widehat{V}_g^{(t)}(\rho)\right)\right] \\
&\quad + \eta_2^2\mathbb{E}\left[\sum_{t=0}^{T-1}(\widehat{V}_g^{(t)}(\rho) - b)^2\right] \\
&\leq 2\eta_2\mathbb{E}\left[\sum_{t=0}^{T-1}\lambda^{(t)}\left(V_g^\star(\rho) - V_g^{(t)}(\rho)\right)\right] + \frac{4\eta_2^2 T}{(1-\gamma)^2}
\end{aligned}
$$

where we use arguments similar to those right below (32): $\mathbb{E}\left[\lambda^{(t)}\left(V_g^{(t)}(\rho) - \widehat{V}_g^{(t)}(\rho)\right)\right] = 0$ and $\mathbb{E}\left[\eta_2^2\sum_{t=0}^{T-1}(\widehat{V}_g^{(t)}(\rho) - b)^2\right] \leq \frac{\eta_2^2 T}{(1-\gamma)^2}\left(2 + \frac{K+1}{K}\right) \leq \frac{4\eta_2^2 T}{(1-\gamma)^2}$.

It should be noticed that $V_r^\star(\rho) = V_r^{\pi_\theta^\star}(\rho)$. Viewing the bound in (30), we return back to

$$
\begin{aligned}
&\frac{1}{T}\sum_{t=0}^{T-1}\lambda^{(t)}\left(V_g^\star(\rho) - V_g^{(t)}(\rho)\right) \\
&\leq \frac{1}{(1-\gamma)\eta_1 T}\sum_{t=0}^{T-1}\left(\mathbb{E}_{s\sim d^\star}\left(D_{\mathrm{KL}}(\pi^\star(\cdot\,|\,s)\,\|\,\pi^{(t)}(\cdot\,|\,s)) - D_{\mathrm{KL}}(\pi^\star(\cdot\,|\,s)\,\|\,\pi^{(t+1)}(\cdot\,|\,s))\right)\right) \\
&\quad + \frac{1}{(1-\gamma)T}\sum_{t=0}^{T-1}\sqrt{E^{\nu^\star}(\widehat{w}^{(t)};\theta^{(t)},\lambda^{(t)})} + \frac{\eta_1\beta\widehat{W}^2}{2(1-\gamma)^3} \\
&\quad - \frac{1}{T}\sum_{t=0}^{T-1}\left(V_r^\star(\rho) - V_r^{(t)}(\rho)\right) \\
&\leq \frac{\log|A|}{(1-\gamma)\eta_1 T} + \frac{1}{(1-\gamma)T}\sum_{t=0}^{T-1}\sqrt{E^{\nu^\star}(\widehat{w}^{(t)};\theta^{(t)},\lambda^{(t)})} + \frac{\eta_1\beta\widehat{W}^2}{2(1-\gamma)^3} + \frac{\lambda^\star + 1}{1-\gamma}
\end{aligned}
$$

where in the last inequality we take telescoping sum in the first sum and drop off a non-positive term; the last sum is due to

$$
\begin{aligned}
V_r^{\pi_\theta^\star}(\rho) &= V_r^\star(\rho) + (V_r^{\pi_\theta^\star}(\rho) - V_r^\star(\rho)) \\
&\geq V_r^\star(\rho) - \frac{1}{1-\gamma} \\
&= V_D^\star(\rho) - \frac{1}{1-\gamma} \\
&= \underset{\pi}{\text{maximize}}\; V_r^\pi(\rho) + \lambda^\star\left(V_g^\pi(\rho) - b\right) - \frac{1}{1-\gamma} \\
&\geq V_r^{(t)}(\rho) + \lambda^\star\left(V_g^{(t)}(\rho) - b\right) - \frac{1}{1-\gamma} \\
&\geq V_r^{(t)}(\rho) - \frac{\lambda^\star + 1}{1-\gamma}
\end{aligned}
$$

where in the second equality we apply the strong duality in Lemma 1, the first and last inequalities are due to the boundedness of $|V_r^{\pi_\theta^\star}(\rho) - V_r^\star(\rho)| \leq \frac{1}{1-\gamma}$ and $|V_g^{(t)}(\rho) - b| \leq \frac{1}{1-\gamma}$. In the above context, we abuse notation $V_r^\star(\rho)$ a bit: $V_r^\star(\rho)$ is described by Lemma 1.

Notice $\left(\mathbb{E}\left[\lambda^{(T)}\right]\right)^2 \leq \mathbb{E}\left[\left(\lambda^{(T)}\right)^2\right]$. Therefore,

$$\frac{1}{\eta_2 T}\,\mathbb{E}\left[\lambda^{(T)}\right] \;\leq\;$$

$$\frac{1}{\eta_2 T}\sqrt{2\eta_2 T\left(\frac{\log|A|}{(1-\gamma)\eta_1 T} + \frac{1}{(1-\gamma)T}\sum_{t=0}^{T-1}\mathbb{E}\left[\sqrt{E^{\nu^\star}(\widehat{w}^{(t)};\theta^{(t)},\lambda^{(t)})}\right] + \frac{\eta_1\beta\widehat{W}^2}{2(1-\gamma)^3} + \frac{\lambda^\star+1}{1-\gamma}\right) + \frac{4\eta_2^2 T}{(1-\gamma)^2}}$$

which leads to the desired bound (28) via (33).

The second part seeks to upper bound the expecation of $E^{\nu^\star}(\widehat{w}^{(t)};\theta^{(t)},\lambda^{(t)})$,

$$\mathbb{E}\left[E^{\nu^\star}(\widehat{w}^{(t)};\theta^{(t)},\lambda^{(t)})\right]$$

$$\leq \quad \mathbb{E}\left[\left\|\frac{\nu^\star}{\nu^{(t)}}\right\|_\infty E^{\nu^{(t)}}(\widehat{w}^{(t)};\theta^{(t)},\lambda^{(t)})\right]$$

$$\leq \quad \frac{1}{1-\gamma}\mathbb{E}\left[\left\|\frac{\nu^\star}{\nu_0}\right\|_\infty E^{\nu^{(t)}}(\widehat{w}^{(t)};\theta^{(t)},\lambda^{(t)})\right]$$

$$= \quad \frac{1}{1-\gamma}\left\|\frac{\nu^\star}{\nu_0}\right\|_\infty\left(\mathbb{E}\left[E_\star^{\nu^{(t)}}(\theta^{(t)},\lambda^{(t)})\right] + \mathbb{E}\left[E^{\nu^{(t)}}(\widehat{w}^{(t)};\theta^{(t)},\lambda^{(t)})\right] - \mathbb{E}\left[E_\star^{\nu^{(t)}}(\theta^{(t)},\lambda^{(t)})\right]\right)$$

$$\leq \quad \frac{1}{1-\gamma}\left\|\frac{\nu^\star}{\nu_0}\right\|_\infty\left(\epsilon_{\text{approx}} + \mathbb{E}\left[E^{\nu^{(t)}}(\widehat{w}^{(t)};\theta^{(t)},\lambda^{(t)})\right] - \mathbb{E}\left[E_\star^{\nu^{(t)}}(\theta^{(t)},\lambda^{(t)})\right]\right).$$

To upper bound $\mathbb{E}\left[E^{\nu^{(t)}}(\widehat{w}^{(t)};\theta^{(t)},\lambda^{(t)})\right] - \mathbb{E}\left[E_\star^{\nu^{(t)}}(\theta^{(t)},\lambda^{(t)})\right]$, we analyze the SGD update in line 9 of Algorithm 1. The SGD update performs minimizing the objective $E^{\nu^{(t)}}(w;\theta^{(t)},\lambda^{(t)})$ with an unbiased estimate of the gradient $\nabla_w E^{\nu^{(t)}}(w_k;\theta^{(t)},\lambda^{(t)})$,

$$\mathbb{E}\left[G_k\,|\,w_k\right]$$

$$= \quad 2\mathbb{E}_{(s,a)\sim\nu^{(t)}}\left[\left(w_k\cdot\nabla_\theta\log\pi^{(t)}(a\,|\,s) - \widehat{A}_L^{(t)}(s,a)\right)\nabla_\theta\log\pi^{(t)}(a\,|\,s)\right]$$

$$= \quad 2\mathbb{E}_{(s,a)\sim\nu^{(t)}}\left[\left(w_k\cdot\nabla_\theta\log\pi^{(t)}(a\,|\,s) - \mathbb{E}\big[\widehat{A}_L^{(t)}(s,a)\,|\,s,a\big]\right)\nabla_\theta\log\pi^{(t)}(a\,|\,s)\right]$$

$$= \quad 2\mathbb{E}_{(s,a)\sim\nu^{(t)}}\left[\left(w_k\cdot\nabla_\theta\log\pi^{(t)}(a\,|\,s) - A_L^{(t)}(s,a)\right)\nabla_\theta\log\pi^{(t)}(a\,|\,s)\right]$$

$$= \quad \nabla_w E^{\nu^{(t)}}(w_k;\theta^{(t)},\lambda^{(t)})$$

where the last equality is due to the fact that: $\widehat{A}_L^{(t)}(s,a)$ is an unbiased estimate of $A_L^{(t)}(s,a)$ in line 8 of Algorithm 1.

By the fast SGD result [5, Theorem 1] with $\alpha = \frac{1}{L_\pi}$ and Assumption 3,

$$\mathbb{E}\left[E^{\nu^{(t)}}(\widehat{w}^{(t)};\theta^{(t)},\lambda^{(t)}) - E_\star^{\nu^t}(\theta^{(t)},\lambda^{(t)})\right] \;\leq\; \frac{2\left(\sigma\sqrt{d} + WL_\pi\right)^2}{K}$$

where $\sigma$ is an uniform bound on the minimum variance,

$$\mathbb{E}_{(s,a)\sim\nu^{(t)}}\left[G_\star^{(t)}\left(G_\star^{(t)}\right)^\top\right] \;\leq\; \sigma^2\,\nabla_w^2 E^{\nu^{(t)}}(w^{(t)};\theta^{(t)},\lambda^{(t)})$$

$$G_\star^{(t)} \;=\; \left(w^{(t)}\cdot\nabla_\theta\log\pi^{(t)}(a\,|\,s) - \widehat{A}_L^{(t)}(s,a)\right)\nabla_\theta\log\pi^{(t)}(a\,|\,s).$$

We complete the second part by noting $\sigma < L_\pi\left\|w^{(t)}\right\| + \frac{1}{1-\gamma} \leq WL_\pi + \frac{1}{1-\gamma}$.

# H  Sample-Based NPG-PD Algorithm with Softmax Parametrization

We describe a sample-based NPG-PD algorithm with softmax parametrization in Algorithm 2. Regarding the computational complexity of Algorithm 2: each round has expected length $2/(1-\gamma)$ so the expected number of total samples is $2(2|S| + |S||A|)KT/(1-\gamma)$; the total number of scalar multiplies, divides, and additions is $O(|S||A|KT + KT/(1-\gamma))$.

---
**Algorithm 2** Sample-Based NPG-PD Algorithm with Softmax Parametrization
---
1: **Initialization**: Learning rates $\eta_1$ and $\eta_2$, number of rounds $K$, and simulation access to CMDP$(S, A, P, r, g, b, \gamma, \rho)$.
2: **for** $t = 0, \ldots, T - 1$ **do**
3:     Initialize $\theta^{(0)} = 0$, $\lambda^{(0)} = 0$, $w_0 = 0$.
4:     Initialize $\widehat{V}_L^{(t)}(s) = 0$ for all $s \in S$ and $\widehat{Q}_L^{(t)}(s, a) = 0$ for all $(s, a) \in S \times A$.
5:     **for** $k = 0, 1, \ldots, K - 1$ **do**
6:         Starting from each $s \in S$, execute policy $\pi^{(t)}$ with a termination probability $1 - \gamma$ and estimate,

$$\widehat{V}_L^{(t)}(s) \;=\; \sum_{k=0}^{K'-1} \Big( r(s_k, a_k) + \lambda^{(t)} g(s_k, a_k) \Big) \quad \text{where } s_0 = s, K' \sim \text{Geo}(1 - \gamma).$$

7:         Starting from each $(s, a) \in S \times A$, execute policy $\pi^{(t)}$ with a termination probability $1 - \gamma$ and estimate,

$$\widehat{Q}_L^{(t)}(s, a) \;=\; \sum_{k=0}^{K'-1} \Big( r(s_k, a_k) + \lambda^{(t)} g(s_k, a_k) \Big) \quad \text{where } s_0 = s, a_0 = a, K' \sim \text{Geo}(1 - \gamma).$$

8:         Update $\widehat{V}_L^{(t)}(s) = \widehat{V}_L^{(t)}(s) + \frac{1}{K} \widehat{V}_L^{(t)}(s)$ for all $s \in S$.
9:         Update $\widehat{Q}_L^{(t)}(s, a) = \widehat{Q}_L^{(t)}(s, a) + \frac{1}{K} \widehat{Q}_L^{(t)}(s, a)$ for all $(s, a) \in S \times A$.
10:     **end for**
11:     Estimate $\widehat{A}_L^{(t)} = \widehat{Q}_L^{(t)} - \widehat{V}_L^{(t)}$.
12:     Initialize $\widehat{V}_g^{(t)}(\rho) = 0$.
13:     **for** $k = 0, 1, \ldots, K - 1$ **do**
14:         Draw $s \sim \rho$ and draw $a \sim \pi^{(t)}(\cdot \,|\, s)$.
15:         Execute policy $\pi^{(t)}$ starting from $s$ with a termination probability $1 - \gamma$ and compute the estimate,

$$\widehat{V}_g^{(t)}(s) \;=\; \sum_{k=0}^{K'-1} g(s_k, a_k) \quad \text{where } s_0 = s, a_0 = a, K' \sim \text{Geo}(1 - \gamma).$$

16:         Update $\widehat{V}_g^{(t)}(\rho) = \widehat{V}_g^{(t)}(\rho) + \frac{1}{K} \widehat{V}_g^{(t)}(s)$.
17:     **end for**
18:     Natural policy gradient primal-dual update

$$
\begin{aligned}
\theta^{(t+1)} &= \theta^{(t)} + \tfrac{\eta_1}{1-\gamma} \widehat{A}_L^{(t)} \\
\lambda^{(t+1)} &= \mathcal{P}_{[0, 1/((1-\gamma)\xi)]} \left( \lambda^{(t)} - \eta_2 \big( \widehat{V}_g^{(t)}(\rho) - b \big) \right).
\end{aligned}
\tag{34}
$$

19: **end for**
---

Similar to Appendix F, we have unbiased estimates,

$$\mathbb{E}\left[\widehat{V}_L^{(t)}(s)\right] = V_L^{(t)}(s) \text{ and } \mathbb{E}\left[\widehat{Q}_L^{(t)}(s,a)\right] = Q_L^{(t)}(s,a) \text{ and } \mathbb{E}\left[\widehat{V}_g^{(t)}(s)\right] = V_g^{(t)}(s)$$

and a variance bound,

$$\mathrm{Var}\left[\widehat{V}_g^{(t)}(s)\right] \leq \frac{1}{(1-\gamma)^2}.$$

# I  Proof of Theorem 4

The proof idea is similar to Theorem 1, we repeat it for readers' convenience.

We highlight some different steps here. The proof is based on similar results as Lemma 6 and Lemma 7 in which essentially we replace the population quantities by the empirical ones estimated by Algorithm 2. It is noted that the trajectory samplings in Algorithm 2 are independent at different times $t$ and all estimates are unbiased. By Lemma 6 and Lemma 7,

$$\mathbb{E}\left[\frac{1}{T}\sum_{t=0}^{T-1}\left(V_r^\star(\rho) - V_r^{(t)}(\rho)\right)\right] + \mathbb{E}\left[\frac{1}{T}\sum_{t=0}^{T-1}\lambda^{(t)}\left(V_g^\star(\rho) - V_g^{(t)}(\rho)\right)\right]$$
$$\leq \frac{\log|A|}{\eta_1 T} + \frac{1}{(1-\gamma)^2 T} + \frac{2\eta_2}{(1-\gamma)^3}. \tag{35}$$

**Bounding the optimality gap**. By the dual update in (34),

$$0 \leq \left(\lambda^{(T)}\right)^2 = \sum_{t=0}^{T-1}\left((\lambda^{(t+1)})^2 - (\lambda^{(t)})^2\right)$$
$$= \sum_{t=0}^{T-1}\left(\left(\mathcal{P}_\Lambda\left(0, \lambda^{(t)} - \eta_2(\widehat{V}_g^{(t)}(\rho) - b)\right)\right)^2 - (\lambda^{(t)})^2\right)$$
$$\leq \sum_{t=0}^{T-1}\left(\left(\lambda^{(t)} - \eta_2(\widehat{V}_g^{(t)}(\rho) - b)\right)^2 - (\lambda^{(t)})^2\right)$$
$$= 2\eta_2\sum_{t=0}^{T-1}\lambda^{(t)}(b - \widehat{V}_g^{(t)}(\rho)) + \eta_2^2\sum_{t=0}^{T-1}(\widehat{V}_g^{(t)}(\rho) - b)^2$$
$$\leq 2\eta_2\sum_{t=0}^{T-1}\lambda^{(t)}\left(V_g^\star(\rho) - V_g^{(t)}(\rho)\right) + 2\eta_2\sum_{t=0}^{T-1}\lambda^{(t)}\left(V_g^{(t)}(\rho) - \widehat{V}_g^{(t)}(\rho)\right)$$
$$+ \eta_2^2\sum_{t=0}^{T-1}(\widehat{V}_g^{(t)}(\rho) - b)^2 \tag{36}$$

where the last inequality is due to the feasibility of the policy $\pi^\star$: $V_g^\star(\rho) \geq b$. Since $V_g^{(t)}(\rho)$ is a population quantity and $\widehat{V}_g^{(t)}(\rho)$ is an estimate that is independent of $\lambda^{(t)}$ given $\theta^{(t-1)}$, $\lambda^{(t)}$ is independent of $V_g^{(t)}(\rho) - \widehat{V}_g^{(t)}(\rho)$ at time $t$ and thus $\mathbb{E}\left[\lambda^{(t)}\left(V_g^{(t)}(\rho) - \widehat{V}_g^{(t)}(\rho)\right)\right] = 0$ due to the fact $\mathbb{E}\left[\widehat{V}_g^{(t)}(\rho)\right] = V_g^{(t)}(\rho)$ (see Appendix H). Therefore,

$$-\mathbb{E}\left[\frac{1}{T}\sum_{t=0}^{T-1}\lambda^{(t)}\left(V_g^\star(\rho) - V_g^{(t)}(\rho)\right)\right] \leq \mathbb{E}\left[\frac{\eta_2}{2T}\sum_{t=0}^{T-1}(\widehat{V}_g^{(t)}(\rho) - b)^2\right]$$
$$\leq \frac{\eta_2}{2(1-\gamma)^2}\left(1 + \frac{K+1}{K}\right)$$

where in the second inequality we drop a non-positive term and use the fact (see Appendix F),

$$\mathbb{E}\left[\widehat{V}_g^{(t)}(\rho)\right] = V_g^{(t)}(\rho)$$

and

$$
\begin{aligned}
\mathbb{E}\left[\left(\widehat{V}_g^{(t)}(\rho)\right)^2\right] &= \frac{1}{K}\mathbb{E}\left[\left(\widehat{V}_g^{(t)}(s)\right)^2\right] + \frac{K-1}{K}\mathbb{E}\left[\widehat{V}_g^{(t)}(s)\right]\mathbb{E}\left[\widehat{V}_g^{(t)}(s)\right] \\
&= \frac{1}{K}\left(\mathrm{Var}\left[\widehat{V}_g^{(t)}(s)\right] + \left(\mathbb{E}\left[\widehat{V}_g^{(t)}(s)\right]\right)^2\right) + \frac{K-1}{K(1-\gamma)^2} \\
&= \frac{1}{K}\left(\mathrm{Var}\left[\widehat{V}_g^{(t)}(s)\right] + \left(V_g^{(t)}(s)\right)^2\right) + \frac{K-1}{K(1-\gamma)^2} \\
&\leq \frac{2}{K(1-\gamma)^2} + \frac{K-1}{K(1-\gamma)^2}
\end{aligned}
$$

where the first equality is due to line 16 of Algorithm 2; the last inequality is due to $\mathrm{Var}\left[\widehat{V}_g^{(t)}(s)\right] \leq \frac{1}{(1-\gamma)^2}$ (see Appendix F) and $0 \leq V_g^{(t)}(s) \leq \frac{1}{1-\gamma}$.

To finish this part, we now return to (35) with $\eta_1 = 2\log|A|$ and $\eta_2 = \frac{1-\gamma}{\sqrt{T}}$,

$$
\mathbb{E}\left[\frac{1}{T}\sum_{t=0}^{T-1}\left(V_r^\star(\rho) - V_r^{(t)}(\rho)\right)\right] \leq \frac{\log|A|}{\eta_1 T} + \frac{1}{(1-\gamma)^2 T} + \frac{2\eta_2}{(1-\gamma)^3} + \frac{\eta_2}{(1-\gamma)^2} + \frac{\eta_2}{2(1-\gamma)^2 K}.
$$

**Bounding the constraint violation**. By the dual update in (34), for any $\lambda \in \left[0, \frac{1}{(1-\gamma)\xi}\right]$,

$$
\begin{aligned}
\mathbb{E}\left[|\lambda^{(t+1)} - \lambda|^2\right] &\\
&= \mathbb{E}\left[\left|\mathcal{P}_\Lambda\left(\lambda^{(t)} - \eta_2\left(\widehat{V}_g^{(t)}(\rho) - b\right)\right) - \mathcal{P}_\Lambda(\lambda)\right|^2\right] \\
&\leq \mathbb{E}\left[\left|\lambda^{(t)} - \eta_2\left(\widehat{V}_g^{(t)}(\rho) - b\right) - \lambda\right|^2\right] \\
&= \mathbb{E}\left[\left|\lambda^{(t)} - \lambda\right|^2\right] - 2\eta_2\mathbb{E}\left[\left(\widehat{V}_g^{(t)}(\rho) - b\right)\left(\lambda^{(t)} - \lambda\right)\right] + \eta_2^2\mathbb{E}\left[\left(\widehat{V}_g^{(t)}(\rho) - b\right)^2\right] \\
&\leq \mathbb{E}\left[\left|\lambda^{(t)} - \lambda\right|^2\right] - 2\eta_2\mathbb{E}\left[\left(\widehat{V}_g^{(t)}(\rho) - b\right)\left(\lambda^{(t)} - \lambda\right)\right] + \frac{2\eta_2^2}{(1-\gamma)^2} + \frac{\eta_2^2}{K(1-\gamma)^2}
\end{aligned}
$$

where the first inequality is due to the non-expansiveness of $\mathcal{P}_\Lambda$ and the last inequality is due to $\mathbb{E}\left[(\widehat{V}_g^{(t)}(\rho) - b)^2\right] \leq \frac{2}{(1-\gamma)^2} + \frac{1}{K(1-\gamma)^2}$. Summing it up from $t = 0$ to $t = T - 1$ and dividing it by $T$ yield

$$
\begin{aligned}
0 &\leq \frac{1}{T}\mathbb{E}\left[|\lambda^{(T)} - \lambda|^2\right] \\
&\leq \frac{1}{T}\mathbb{E}\left[\left|\lambda^{(0)} - \lambda\right|^2\right] - \frac{2\eta_2}{T}\sum_{t=0}^{T-1}\mathbb{E}\left[\left(\widehat{V}_g^{(t)}(\rho) - b\right)\left(\lambda^{(t)} - \lambda\right)\right] + \frac{2\eta_2^2}{(1-\gamma)^2} + \frac{\eta_2^2}{K(1-\gamma)^2}
\end{aligned}
$$

which further implies,

$$
\mathbb{E}\left[\frac{1}{T}\sum_{t=0}^{T-1}\left(V_g^{(t)}(\rho) - b\right)\left(\lambda^{(t)} - \lambda\right)\right] \leq \frac{1}{2\eta_2 T}\mathbb{E}\left[\left|\lambda^{(0)} - \lambda\right|^2\right] + \frac{\eta_2}{(1-\gamma)^2} + \frac{\eta_2}{2K(1-\gamma)^2}
$$

where we use $\mathbb{E}\left[\widehat{V}_g^{(t)}(\rho)\right] = V_g^{(t)}(\rho)$ and $\lambda^{(t)}$ is independent of $\widehat{V}_g^{(t)}(\rho)$ given $\theta^{(t-1)}$. We now add the above inequality into (35) and note $V_g^\star(\rho) \geq b$,

$$
\begin{aligned}
&\mathbb{E}\left[\frac{1}{T}\sum_{t=0}^{T-1}\left(V_r^\star(\rho) - V_r^{(t)}(\rho)\right)\right] + \lambda\mathbb{E}\left[\frac{1}{T}\sum_{t=0}^{T-1}\left(b - V_g^{(t)}(\rho)\right)\right] \\
&\leq \frac{\log|A|}{\eta_1 T} + \frac{1}{(1-\gamma)^2 T} + \frac{2\eta_2}{(1-\gamma)^3} + \frac{1}{2\eta_2 T}\left|\lambda^{(0)} - \lambda\right|^2 + \frac{\eta_2}{(1-\gamma)^2} + \frac{\eta_2}{2K(1-\gamma)^2}.
\end{aligned}
$$

We take $\lambda = \frac{2}{(1-\gamma)\xi}$ when $\sum_{t=0}^{T-1}\left(b - V_g^{(t)}(\rho)\right) \geq 0$; otherwise $\lambda = 0$. Thus,

$$
\begin{aligned}
&\mathbb{E}\left[V_r^\star(\rho) - \frac{1}{T}\sum_{t=0}^{T-1}V_r^{(t)}(\rho)\right] + \frac{2}{(1-\gamma)\xi}\mathbb{E}\left[b - \frac{1}{T}\sum_{t=0}^{T-1}V_g^{(t)}(\rho)\right]_+ \\
&\leq \frac{\log|A|}{\eta_1 T} + \frac{1}{(1-\gamma)^2 T} + \frac{2\eta_2}{(1-\gamma)^3} + \frac{1}{2\eta_2(1-\gamma)^2\xi^2 T} + \frac{\eta_2}{(1-\gamma)^2} + \frac{\eta_2}{2K(1-\gamma)^2}.
\end{aligned}
$$

Similar to the proof of Theorem 1, there exists a policy $\pi'$ such that $V_r^{\pi'}(\rho) = \frac{1}{T}\sum_{t=0}^{T-1} V_r^{(t)}(\rho)$ and $V_g^{\pi'}(\rho) = \frac{1}{T}\sum_{t=0}^{T-1} V_g^{(t)}(\rho)$. Therefore,

$$\mathbb{E}\left[V_r^{\star}(\rho) - V_r^{\pi'}(\rho)\right] + \frac{2}{(1-\gamma)\xi}\mathbb{E}\left[b - V_g^{\pi'}(\rho)\right]_+$$
$$\leq \frac{\log|A|}{\eta_1 T} + \frac{1}{(1-\gamma)^2 T} + \frac{2\eta_2}{(1-\gamma)^3} + \frac{1}{2\eta_2(1-\gamma)^2\xi^2 T} + \frac{\eta_2}{(1-\gamma)^2} + \frac{\eta_2}{2K(1-\gamma)^2}.$$

According to Theorem 6 in Appendix C, we obtain

$$\mathbb{E}\left[b - V_g^{\pi'}(\rho)\right]_+ \leq \frac{\xi\log|A|}{\eta_1 T} + \frac{\xi}{(1-\gamma)T} + \frac{2\eta_2\xi}{(1-\gamma)^2} + \frac{1}{2\eta_2(1-\gamma)\xi T} + \frac{\eta_2\xi}{(1-\gamma)} + \frac{\eta_2\xi}{2K(1-\gamma)}$$

which shows the constraint violation bound by noting $\frac{1}{T}\sum_{t=0}^{T-1}\left(b - V_g^{(t)}(\rho)\right) = b - V_g^{\pi'}(\rho)$ and taking $\eta_1 = 2\log|A|$ and $\eta_2 = \frac{1-\gamma}{\sqrt{T}}$.

## J Experimental Results

In this section, we provide additional experimental results to support our convergence theory. Our CMDP simulation is based on the shared MDP code [9]. We generate CMDPs with random transitions, uniform rewards, and utilities in $[0,1]$. We simulate our algorithms with random initializations. Given $T > 0$, the total number of optimization iterations, our stepsizes in theorems become constants and multiplying them with positive constants does not affect convergence rates.

Figure 4: Convergence of the NPG-PD method (10). In this experiment, we have randomly generated a CMDP with $|S| = 20$, $|A| = 10$, $\gamma = 0.8$, and $b = 3$, and chosen: $\eta_1 = \eta_2 = 0.1$ and $d = 150$.

We show simulation results for algorithms with the general smooth parametrization. We consider a class of linear softmax policies,

$$\pi_\theta(a \,|\, s) = \frac{\exp(\theta \cdot \phi_{s,a})}{\sum_{a' \in A}\exp(\theta \cdot \phi_{s',a'})}$$

where $\phi_{s,a} \in \mathbb{R}^d$ is the feature map with $\|\phi_{s,a}\| \leq \beta$. We compute $\nabla_\theta \log \pi_\theta(a \,|\, s) = \phi_{s,a} - \mathbb{E}_{a' \sim \pi_\theta(\cdot \,|\, s)}[\phi_{s,a'}] := \widetilde{\phi}_{s,a}$ and the compatible function approximation error,

$$E^\nu(w; \theta, \lambda) = \mathbb{E}_{s,a \sim \nu}\left[\left(A_L^{\pi_\theta,\lambda}(s,a) - w \cdot \widetilde{\phi}_{s,a}\right)^2\right].$$

In this experiment, we take $d$ canonical bases in $\mathbb{R}^d$ as our feature maps. Since $d < |S||A|$, they can't capture the advantage function and will introduce function approximation errors. In Figure 4, we only show the convergence of the reward value function to a stationary value that could be sub-optimal due to the function approximation error. By contrast, the constraint violation converges to zero sublinearly. It verifies Theorem 2 that the function approximation error does not dominate the constraint violation.

Last but not least, we show the objective and the constraint violation for running the sample-based NPG-PD algorithm (10): Algorithm 1, using two different sample sizes. We see that both reward value functions converge, and both constraint violations decrease to be negative. The large sample size of $K = 200$ performs better, especially for the constraint violation. It confirms Theorem 3 that the constraint violation is insusceptible to the function approximation error.

Figure 5: Convergence of the sample-based NPG-PD algorithm (10): Algorithm 1, using different sample sizes: $K = 100$ (- -) and $K = 200$ (—). In this experiment, we have randomly generated a CMDP with $|S| = 20$, $|A| = 10$, $\gamma = 0.8$, and $b = 3$. We have chosen parameters for Algorithm 1: $\eta_1 = \eta_2 = 0.1$, $\alpha = 0.1$, and $d = 150$.