[Reviews · NeurIPS 2020]

Review 1

Summary and Contributions: The paper formulates natural-gradient-based primal-dual methods to solve CMDP and show global convergence of the method. The global convergence property is shown to hold on nonconcave maximization problems with nonconvex constraint sets with parametrized settings. With the softmax policy that is complete, the sample complexity is also shown to be dimension-free.

Strengths: Related work and references throughout the paper show the author is knowledgeable in the field.

Weaknesses: Major concerns: The contribution of this paper is somewhat trivial, and the generalizability of the analysis method, and the result remains unclear. For example, the NPD method the author proposes must use a natural gradient for the primal and projected subgradient for the dual. The subgradient method is known to be very slow in practice. There are certain places that are not clearly identified. For example, the algorithm needs to use a projection for the dual variable. As a result, the solution gained may not even be the stationary point of the original problem due to the dual projection. How this problem is addressed in not quite clear in the paper. The original problem formulation (Eq (1)) is mentioned to be non-concave as well. How does the Slater condition still ensure strong duality? The proposition mentioned in Lemma 2 from another work seems to be interpreted in a few ways in the paper, and it wasn’t clear to me how it affected strong duality. Settings of \eta’s in the experiments appear to be independent of the theorems. Does the softmax algorithm still achieve “dimension-free” global convergence in spite of it? Minor concerns: Some assumptions are missing. The underlying assumption of the ergodicity is needed because the existence of the stationary distribution is needed in some of the proofs. Line 86: additional effort is *required*

Correctness: As stated above, there are some major technical issues in which I have doubts.

Clarity: The paper is well-written.

Relation to Prior Work: It is clearly discussed how this work differs from previous contributions.

Reproducibility: No

Additional Feedback:


Review 2

Summary and Contributions: In this work, the authors propose a Natural Primal-Dual Policy Gradient method for CMDPs and analyze the global convergence properties with softmax parameterized policies and general policy classes. They show that the NPD method has sublinear rates w.r.t. to the optimal return and constraint satisfaction up to a function approximation error. They also provide the sample complexity analysis of their methods for both the policy class settings and provide empirical results to demonstrate the validity of their approach.

Strengths: The paper is well written and easy to follow, which is upfront about the assumptions they make and the contributions they make. They are the first ones (to my knowledge) to have non-asymptotic theoretical guarantees on the PG methods for CMDPs, even though they are heavily employing the techniques from [1, 2]. The results make sense and most of the assumptions employed are standard in the community. [1] Constrained reinforcement learning has zero duality gap - Paternain et al, 2019 [2] Optimality and approximation with policy gradient methods in Markov decision processes - Agarwal et al, 2019

Weaknesses: - While the analysis of the softmax parameterized policies is appreciated, however in practice when the |S||A| is small enough to allow to work with this, it is also possible to employ LP-based solvers to work directly with the dual formation (Altman 1999). Without addressing why this particular class of policies is of interest in comparison to the Dual-LP approach, the results of the softmax parameterized policies lose some of their weight. I hope either the authors can address the reason to work with such policy class or emphasize more on the general policy class parts of their work. - With regards to Assumption 1, one of the critiques of the original work (Paternain et al) is the discrepancy between employing the slater's condition to show strong-duality when the constraints are non-convex in the first place. I think the authors address that in Theorem 6, but it'll nice to bring that to the main text also. - Is there a reason why the discounted sum wasn't taken for the Algorithms and instead a termination probability of (1 - \gamma) was uesd?

Correctness: The methods seems technically sound.

Clarity: Yes.

Relation to Prior Work: Yes.

Reproducibility: Yes

Additional Feedback:


Review 3

Summary and Contributions: This paper provides a convergence analysis in the constrained MDP framework based on natural gradients with softmax policies. It builds from several recent works demonstrating convergence for policy gradient/optimization methods, and extends it to the CMDP framework with analysis of constraint violation. Their convergence result establishes sublinear rate of convergence even in presence of function approximation with restricted policy classes. It provides a significant contribution for the RL theory community, especially establishing how the convergence analysis differs for primal-dual methods within the CMDP framework. Such a paper will be of significant importance to the RL community.

Strengths: Comments about the paper : This paper presents convergence analysis of primal-dual natural policy gradient methods under the CMDP framework. Several recent works have shown convergence of policy gradients and optimality bounds (e.g Agarwal et al., Mei et al), but the paper extends similar analysis to (a) natural policy gradients (b) CMDP framework with constraints. Overall, it archives a sublinear rate of convergence in the CMDP framework, similar to other related works with convergence analysis. The analysis of the paper is done for the general MDP case with function approximation and restricted policy classes. It is a very well written paper that is easy to follow with significant theoretical derivation and proof details. Supplementary material includes proof of all theorems and lemmas (for completeness as well), which makes the paper quite easy to follow. Lemma 3 provides a nice analysis with a toy example, establishing why the primal and dual objectives are non-concave and non-convex. It then considers both softmax and restricted policy classes for analysis, with a short detail of how natural policy gradient algorithm differs in the constrained MDP case, ie, given constraints based on episodic returns and value functions (V_g), the derivation only differs in the advantage function evaluation step. Lemma 4 is exploited by several prior works as well, establishing the closed form solution of softmax policies. The authors extend this for advantage function with constraints, and shows that similar closed form expression holds (equation 8), which is actually key to the analysis later, for showing performance improvement bounds and the optimality gap. Theorem 1 establishes the key result of the paper, showing 1 / sqrt(T) dependency of the rate of convergence of the primal (optimality gap) and the dual (constraint violation). Appendix D contains detailed proof. Lemma 6 and 7 are easy to follow, and theoretically looks correct. Theorem 2 further extends the convergence analysis for general policy classes, although interestingly now it depends on the distribution ratio coefficient (similarly introduced by Agarwal et al). The constraint violation analysis in both theorem 1 and 2 would actually be interesting for the theoretical safe RL community, since there isn’t much work on constraint violation analysis and establishing a bound in such cases. Sample complexity results are definitely interesting and is a significant novel contribution of this paper. Even though prior works only establish a convergence rate, this paper is quite useful due to this sample complexity result, previously not so established for policy gradient convergence papers. Overall, I definitely recommend this paper for acceptance, and I think it would be of significant impact to the RL theory community. The theoretical analysis of this paper has a significantly novel contribution, well written and demonstrated, such as to be ready for acceptance at NeurIPS.

Weaknesses: Few comments for improving the paper : Theorem 2 depends on the distribution ratio coefficient compared to theorem 1 - either in the main paper, or in the appendix, it might be useful to include details, or short description, as to what the key difference is, that leads to the general policy class case to be dependent on this mismatch coefficient? For example, parallel to this work (on arxiv : [1] Fast Global Convergence of Natural Policy Gradient Methods with Entropy Regularization) establishes that even for natural gradients in the general MDP case, the optimality gap depends on the distribution mismatch coefficient. It seems surprising that theorem 1 is independent of it, while the resulting bound for theorem 2 depends on it? Assumption 2 on the policy smoothness : Is there a detailed derivation for this assumption 2 for the smoothness guarantee? As in the paper above [1], and in (Mae et al., ICML’20) it might be useful to establish a proof for this smoothness result for the general policy classes (since prior works only demonstrate it for softmax policies). Similar to (Mae et al., 2020), it might be useful to establish the non-uniform PL condition for the gradient of the value function (with constraints) in this paper. It might be interesting to see whether similar convergence bounds can be obtained, following a similar type of analysis as in Mae et al.

Correctness: The paper has significant theoretical contributions. I have carefully checked the appendix of the paper as well, and it has a solid contribution with details that are easy to follow.

Clarity: The paper is extremely well written and easy to follow, and shows a step by step proof of convergence bound, while also relating to other prior works demonstrating similar results. Few results are also included in the appendix for completeness.

Relation to Prior Work: Prior works are detailed in the main paper, and I myself am familiar with a lot of prior works, that were also included in the related works section.

Reproducibility: Yes

Additional Feedback:


Review 4

Summary and Contributions: This paper focus on establishing theoretical guarantees for solving constrained Markov Decision Process (MDP) using Primal-Dual natural Policy Gradient methods. With a non-concave objective function and non-convex constraint set, the authors are able to prove global convergence rate for two different parametrization (i.e. softmax and general). Two sampling based methods are provided with sample complexity guarantees.

Strengths: This work improves upon previous theoretical analysis on solving constrained MDP using Primal Dual natural Policy Gradient method. For softmax parametrization, dimensional-free global convergence is achieved following previous dual analysis. For general parameterization, the optimality and feasibility gap exhibits different convergence rate, which is worth further study.

Weaknesses: In the discussion of the general class of policies. The authors stated that if the dimensionality of parameter space is much smaller than the state-action space, the policy class has limited expressiveness. Can the authors explain intuitively why would lead to global convergence? In the theorems, many parameters or intrinsic quantities tend to influence the convergence rate. Can the authors provide a plot about the empirical and theoretical result comparison? For example, the sample amount K and the bias epsilon_approx are theoretically shown to be dominating factors of the convergence. In general, any sensible gap between the theoretical and empirical results should be inspiring. Another interesting fact about the different dependency of optimality gap and feasibility gap is less discussed. More comprehensive explanation or empirical evaluation can lead to more impact to the readers.

Correctness: Yes.

Clarity: Yes.

Relation to Prior Work: Yes.

Reproducibility: Yes

Additional Feedback:

[Author Response · NeurIPS 2020]

**R#1**: Thank you for your comments. We believe that we have addressed all of your concerns and hope that you would be open to re-evaluating quality of our contribution.

(1). *somewhat trivial*, *remains unclear*: See the review from R#3 regarding our contributions/strengths. In particular, (i) Compared to minimax optimization, e.g., [24], we exploit the problem structure (e.g., Lemma 4) to prove global convergence and incorporate function approximation error in Section 5. This approach has not been previously utilized in RL; (ii) We are the first to establish global convergence rate. Our guarantees are much stronger than the folklore stationary-point results [34, 25] and the rate for the softmax case matches the best-known one [27, 45]; (iii) Addressing the following challenges required the development of new methods that are far from being trivial: lack of monotonic improvement [3], constraint coupling, losing strong duality for the general case, and estimating gradients via samples.

(2). *natural gradient for the primal*: Our natural primal-dual method employs a *first principle thinking* to solve saddle-point problems; see, e.g., [R1]. The primal update has nice features: (i) It is well-known that NPG is closely related to popular RL algorithms, e.g., TRPO [36] and PPO [37]. Our method enables extending such RL algorithms to constrained problems; (ii) In the softmax case, the update has a concise form with multiplicative weights, which is free of any state distributions because of Fisher preconditioning; (iii) In the general case, the update naturally takes the compatible function approximation [3, 19], thereby enabling approximation of the primal update via regression with samples.

(3). *projection for the dual variable*: Since we ensure the optimal dual variable $\lambda^\star$ to be in the projection interval $\Lambda$ (see Lemma 2), the projection does not impact the constraint violation guarantees. Such a projection is commonly used and it can be found in [R1] for constrained convex optimization and in [15] for CMDPs in standard LP forms. In (9b), $[\cdot]_+$ reflects the left bound of $\Lambda$ and the right one is captured via $\xi$. Since the right bound of $\Lambda$ is a constant and $\lambda^\star \in \Lambda$, the violation in Theorem 1 still vanishes sublinearly. In Theorem 2, the projection is only weakened to the left side.

(4). *Slater condition –> strong duality*: Proof of strong duality for (1) using the Slater condition can be found in [34]. In Lemma 2, we prove that the optimal dual variable is bounded under the Slater condition. These properties for CMDPs are similar to familiar properties in constrained convex optimization [R1, 7].

(5). *$\eta$'s*, *dimension-free*: They are dependent. Given $T > 0$, we use $\eta$'s in the theorems as multiplying positive constants. This choice does not affect the dimension-free rate and a wide range of stepsizes appears to work in experiments.

(6). *ergodicity*: We use it implicitly as in the PG literature [3, 9] which is standard. We will add clarifications.

**R#2**: Thank you for your positive comments and constructive suggestions.

(1). *softmax parameterized polices*: Reasons using the softmax class: (i) it served as a warm-up for studying RL algorithms; e.g., see [3, 28, R2] and it provided a lens to interpret the compatible function approximation error [3]; (ii) it has nice analytical properties: completeness and differentiability; (iii) it induces a natural update; see (2)(ii, iii) to R#1.

(2). *Dual-LP*: Reasons why PG method avails Dual-LP: (i) it's simple to apply with theoretical guarantees, e.g., [3, 9]; (ii) it's easy to deal with large state-action spaces via policy parametrization, e.g., neural nets [36, 25]; (iii) it directly optimizes/targets the value functions of interest; (iv) it's handy for estimating gradients via simulations of the policy.

(3). *Theorem 6 –> the main text*, *discounted sum*: We will add Theorem 6 before Theorem 1. Randomly terminating with probability $1 - \gamma$ gives an unbiased estimate of the infinite discounted reward; see Appendix F or [3, 48] for proof.

**R#3**: Thank you for recognizing the contributions/strengths of our paper and for providing valuable comments.

(1). *distribution mismatch*: It is expected: Theorem 1 is free of distribution mismatch ratios because any state distribution shifts are canceled by Fisher preconditioning; see Lemma 4 or NPG [3]. In contrast, such a cancellation fails for PG [28] or regularized NPG [R2]. Theorem 2 generalizes NPG via compatible function approximation (FA) that yields a state-action distribution shift. Non-zero FA errors lead to a natural dependence on distribution ratios.

(2). *policy smoothness*: Assumption 2 also holds for the linear softmax policy with bounded feature mappings.

(3). *non-uniform PL*: Our NPG results are independent of the non-uniform PL in PG analysis [28]. We agree with your assessment that it is useful and promising. Our ongoing work proves the convergence of PG for CMDPs in LP forms.

**R#4**: Thank you for your positive comments and insightful questions.

(1). *global convergence*: When the policy class has limited expressiveness, Theorem 2 establishes convergence up to a function approximation (FA) error. Such an error reflects the expressiveness of the parametric policy class. When the parametrization is rich enough, e.g., tabular softmax policy, the associated FA error can be zero, and Theorem 2 establishes the usual 'global convergence.' Even though FA errors enter into the bounds, our theory is still stronger than the folklore stationary-point results in the FA setting; see, e.g., [25].

(2). *empirical and theoretical result comparison*: We provide plots for comparison in Appendix J. Sample-based algorithms converge slower, i.e., more gradient evaluations are needed to reach the same accuracy or stationarity. This verifies the effect of parameters $K$ and $\epsilon_{\mathrm{approx}}$ in Theorems 3 and 4 and yields loose bounds on optimality/violation gaps.

(3). *different dependency*: In contrast to optimality gaps in Theorems 2 and 3, the effect of $\epsilon_{\mathrm{approx}}$ vanishes in constraint violations for a large value of $T$. In Appendix J, we also observe that constraint violations usually converge quickly. We will add this observation in the revised version and provide explanations on other dependencies.

[R1] Nedić, A. and Ozdaglar, A. "Subgradient methods for saddle-point problems." *JOTA*. (2009).

[R2] Cen, S., Cheng, C., Chen, Y., Wei, Y. and Chi, Y. "Fast global convergence of natural policy gradient methods with entropy regularization." *arXiv preprint arXiv:2007.06558*. (2020).


[Meta-Review · NeurIPS 2020]

After reading the authors' rebuttal, the reviewers discussed their concerns about this paper. Ultimately, a consensus was not reached asreviewer #1 feels that the issues raised in her/his review were not properly addressed in the authors' feedback. The other reviewers also share some of the concerns raised by reviewer #1, but, given the rebuttals, they believe the authors can fix them in the final version and make the contribution of their paper clearer. I agree with them and so I suggest to accept the paper, but I recommend that the authors take into consideration the issues raised in the reviews and address them carefully in the final version of the paper.